# EGSDE: Unpaired Image-to-Image Translation via Energy-Guided Stochastic Differential Equations

**Min Zhao[1], Fan Bao[1], Chongxuan Li[2,3*], Jun Zhu[1*]**

[1]Dept. of Comp. Sci. & Tech., BNRist Center, THU-Bosch ML Center, Tsinghua University, China
[2] Gaoling School of Artificial Intelligence, Renmin University of China, Beijing, China
[3] Beijing Key Laboratory of Big Data Management and Analysis Methods , Beijing, China
[4] Pazhou Laboratory (Huangpu), Guangzhou, China
gracezhao1997@gmail.com; bf19@mails.tsinghua.edu.cn;
chongxuanli@ruc.edu.cn; dcszj@tsinghua.edu.cn

## Abstract

Score-based diffusion models (SBDMs) have achieved the SOTA FID results in unpaired image-to-image translation (I2I). However, we notice that existing methods totally ignore the training data in the source domain, leading to sub-optimal solutions for unpaired I2I. To this end, we propose energy-guided stochastic differential equations (EGSDE) that employs an energy function pretrained on both the source and target domains to guide the inference process of a pretrained SDE for realistic and faithful unpaired I2I. Building upon two feature extractors, we carefully design the energy function such that it encourages the transferred image to preserve the domain-independent features and discard domain-specific ones. Further, we provide an alternative explanation of the EGSDE as a product of experts, where each of the three experts (corresponding to the SDE and two feature extractors) solely contributes to faithfulness or realism. Empirically, we compare EGSDE to a large family of baselines on three widely-adopted unpaired I2I tasks under four metrics. EGSDE not only consistently outperforms existing SBDMs-based methods in almost all settings but also achieves the SOTA realism results without harming the faithful performance. Furthermore, EGSDE allows for flexible trade-offs between realism and faithfulness and we improve the realism results further (e.g., FID of 51.04 in Cat $\rightarrow$ Dog and FID of 50.43 in Wild $\rightarrow$ Dog on AFHQ) by tuning hyper-parameters. The code is available at https://github.com/ML-GSAI/EGSDE.

## 1   Introduction

Unpaired image-to-image translation (I2I) aims to transfer an image from a source domain to a related target domain, which involves a wide range of computer vision tasks such as style transfer, super-resolution and pose estimation [35]. In I2I, the translated image should be *realistic* to fit the style of the target domain by changing the domain-specific features accordingly, and *faithful* to preserve the domain-independent features of the source image. Over the past few years, generative adversarial networks [12] (GANs)-based methods [10, 60, 54, 36, 3, 57, 44, 19, 17, 26, 10] dominated this field due to their ability to generate high-quality samples.

In contrast to GANs, score-based diffusion models (SBDMs) [48, 16, 34, 49, 2, 31] perturb data to a Gaussian noise by a diffusion process and learn the reverse process to transform the noise back to the data distribution. Recently, SBDMs achieved competitive or even superior image generation performance to GANs [9] and thus were naturally applied to unpaired I2I [7, 32], which have achieved

---

*Correspondence to Chongxuan Li and Jun Zhu.

36th Conference on Neural Information Processing Systems (NeurIPS 2022).

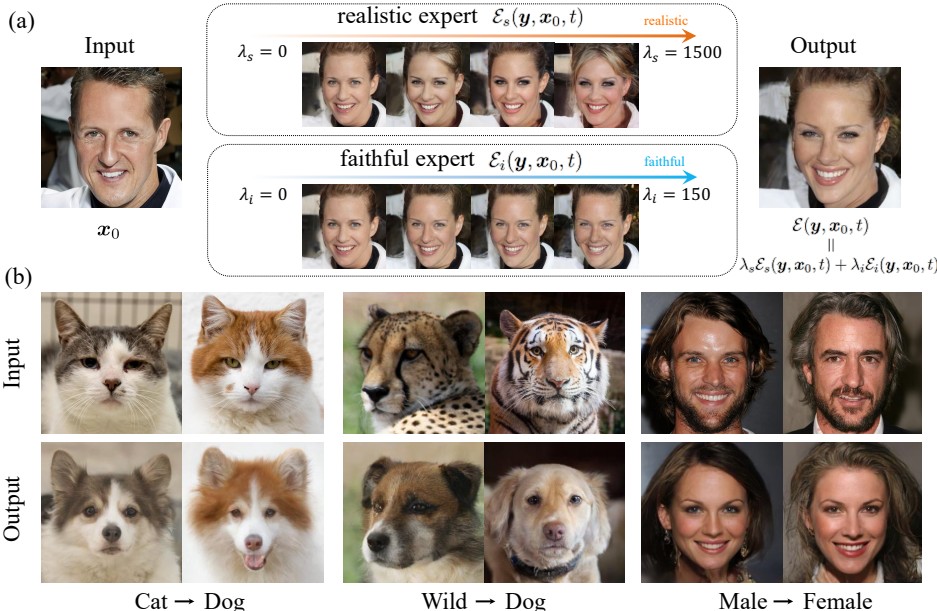

Figure 1: (a) Apart from the SDE, the EGSDE incorporates a realism expert and a faithful expert to preserve the domain-independent features and discard domain-specific ones. (b) Representative translation results on three unpaired I2I tasks.

the state-of-the-art FID [13] and KID [4] results empirically. However, we notice that these methods *did not leverage the training data in the source domain at all*. Indeed, they trained a diffusion model solely on the target domain and exploited the test source image during inference (see details in Sec. 2.2). Therefore, we argue that if the training data in the source domain can be exploited together with those in the target domain, one can learn domain-specific and domain-independent features to improve both the realism and faithfulness of the SBDMs in unpaired I2I.

To this end, we propose *energy-guided stochastic differential equations* (EGSDE) that employs an energy function pretrained across the two domains to guide the inference process of a pretrained SDE for realistic and faithful unpaired I2I. Formally, EGSDE defines a valid conditional distribution via a reverse time SDE that composites the energy function and the pretrained SDE. Ideally, the energy function should encourage the transferred image to preserve the domain-independent features and discard domain-specific ones. To achieve this, we introduce two feature extractors that learn domain-independent features and domain-specific ones respectively, and define the energy function upon the similarities between the features extracted from the transferred image and the test source image. Further, we provide an alternative explanation of the discretization of EGSDE in the formulation of *product of experts* [15]. In particular, the pretrained SDE and the two feature extractors in the energy function correspond to three experts and each solely contributes to faithfulness or realism.

Empirically, we validate our method on the widely-adopted AFHQ [8] and CelebA-HQ [20] datasets including Cat → Dog, Wild → Dog and Male → Female tasks. We compare to a large family of baselines, including the GANs-based ones [36, 60, 17, 26, 3, 10, 57, 58] and SBDMs-based ones [7, 32] under four metrics (e.g., FID). EGSDE not only consistently outperforms SBDMs-based methods in almost all settings but also achieves the SOTA realism results without harming the faithful performance. Furthermore, EGSDE allows for flexible trade-offs between realism and faithfulness and we improve the FID further (e.g., 51.04 in Cat → Dog and 50.43 in Wild → Dog ) by tuning hyper-parameters. EGSDE can also be extended to multi-domain translation easily.

## 2 Background

### 2.1 Score-based Diffusion Models

Score-based diffusion models (SBDMs) gradually perturb data by a forward diffusion process, and then reverse it to recover the data [49, 2, 47, 16, 9]. Let $q(\boldsymbol{y}_0)$ be the unknown data distribution

on $\mathbb{R}^D$. The forward diffusion process $\{y_t\}_{t \in [0,T]}$, indexed by time $t$, can be represented by the following forward SDE:

$$d\boldsymbol{y} = \boldsymbol{f}(\boldsymbol{y}, t)dt + g(t)d\boldsymbol{w}, \tag{1}$$

where $\boldsymbol{w} \in \mathbb{R}^D$ is a standard Wiener process, $\boldsymbol{f}(\cdot, t) : \mathbb{R}^D \to \mathbb{R}^D$ is the drift coefficient and $g(t) \in \mathbb{R}$ is the diffusion coefficient. The $f(\boldsymbol{y}, t)$ and $g(t)$ is related into the noise size and determines the perturbation kernel $q_{t|0}(\boldsymbol{y}_t|\boldsymbol{y}_0)$ from time 0 to $t$. In practice, the $f(\boldsymbol{y}, t)$ is usually affine so that the the perturbation kernel is a linear Gaussian distribution and can be sampled in one step.

Let $q_t(\boldsymbol{y})$ be the marginal distribution of the SDE at time $t$ in Eq. (1). Its time reversal can be described by another SDE [49]:

$$\mathrm{d}\boldsymbol{y} = [\boldsymbol{f}(\boldsymbol{y}, t) - g(t)^2 \nabla_{\boldsymbol{y}} \log q_t(\boldsymbol{y})]\mathrm{d}t + g(t)\mathrm{d}\overline{\boldsymbol{w}}, \tag{2}$$

where $\overline{\boldsymbol{w}}$ is a reverse-time standard Wiener process, and $\mathrm{d}t$ is an infinitesimal negative timestep. [49] adopts a score-based model $\boldsymbol{s}(\boldsymbol{y}, t)$ to approximate the unknown $\nabla_{\boldsymbol{y}} \log q_t(\boldsymbol{y})$ by score matching, thus inducing a score-based diffusion model (SBDM), which is defined by a SDE:

$$\mathrm{d}\boldsymbol{y} = [\boldsymbol{f}(\boldsymbol{y}, t) - g(t)^2 \boldsymbol{s}(\boldsymbol{y}, t)]\mathrm{d}t + g(t)\mathrm{d}\overline{\boldsymbol{w}}. \tag{3}$$

There are numerous SDE solver to solve the Eq. (3) to generate images. [49] discretizes it using the Euler-Maruyama solver. Formally, adopting a step size of $h$, the iteration rule from $s$ to $t = s - h$ is:

$$\boldsymbol{y}_t = \boldsymbol{y}_s - [\boldsymbol{f}(\boldsymbol{y}_s, s) - g(s)^2 \boldsymbol{s}(\boldsymbol{y}_s, s)]h + g(s)\sqrt{h}\boldsymbol{z}, \quad \boldsymbol{z} \sim \mathcal{N}(\boldsymbol{0}, \boldsymbol{I}). \tag{4}$$

## 2.2 SBDMs in Unpaired Image to Image Translation

Given unpaired images from the source domain $\mathcal{X} \subset \mathbb{R}^D$ and the target domain $\mathcal{Y} \subset \mathbb{R}^D$ as the training data, the goal of unpaired I2I is to transfer an image from the source domain to the target domain. Such a process can be formulated as designing a distribution $p(\boldsymbol{y}_0|\boldsymbol{x}_0)$ on the target domain $\mathcal{Y}$ conditioned on an image $\boldsymbol{x}_0 \in \mathcal{X}$ to transfer. The translated image should be *realistic* for the target domain by changing the domain-specific features and *faithful* for the source image by preserving the domain-independent features.

ILVR [7] uses a diffusion model on the target domain for realism. Formally, ILVR starts from $\boldsymbol{y}_T \sim \mathcal{N}(\boldsymbol{0}, \boldsymbol{I})$ and samples from the diffusion model according to Eq. (4) to obtain $\boldsymbol{y}_t$. For faithfulness, it further refines $\boldsymbol{y}_t$ by adding the residual between the sample $\boldsymbol{y}_t$ and the perturbed source image $\boldsymbol{x}_t$ through a non-trainable low-pass filter

$$\boldsymbol{y}_t \leftarrow \boldsymbol{y}_t + \Phi(\boldsymbol{x}_t) - \Phi(\boldsymbol{y}_t), \quad \boldsymbol{x}_t \sim q_{t|0}(\boldsymbol{x}_t|\boldsymbol{x}_0), \tag{5}$$

where $\Phi(\cdot)$ is a low-pass filter and $q_{t|0}(\cdot|\cdot)$ is the perturbation kernel determined by the forward SDE in Eq. (1).

Similarly, SDEdit [32] also adopts a SBDM on the target domain for realism, i.e., sampling from the SBDM according to Eq. (4). For faithfulness, SDEdit starts the generation process from the noisy source image $\boldsymbol{y}_M \sim q_{M|0}(\boldsymbol{y}_M|\boldsymbol{x}_0)$, where $M$ is a middle time between 0 and $T$, and is chosen to preserve the original overall structure and discard local details. We use $p_{r1}(\boldsymbol{y}_0|\boldsymbol{x}_0)$ to denote the marginal distribution defined by such SDE conditioned on $\boldsymbol{x}_0$.

Notably, these methods did not leverage the training data in the source domain at all and thus can be sub-optimal in terms of both the realism and faithfulness in unpaired I2I.

# 3 Method

To overcome the limitations of existing methods [7, 32] as highlighted in Sec. 2.2, we propose *energy-guided stochastic differential equations* (EGSDE) that employs an energy function pre-trained across the two domains to guide the inference process of a pretrained SDE for realistic and faithful unpaired I2I (see Fig. 2). EGSDE defines a valid conditional distribution $p(\boldsymbol{y}_0|\boldsymbol{x}_0)$ by compositing a pretrained SDE and a pretrained energy function under mild regularity conditions[2] as follows:

$$\mathrm{d}\boldsymbol{y} = [\boldsymbol{f}(\boldsymbol{y}, t) - g(t)^2(\boldsymbol{s}(\boldsymbol{y}, t) - \nabla_{\boldsymbol{y}}\mathcal{E}(\boldsymbol{y}, \boldsymbol{x}_0, t))]\mathrm{d}t + g(t)\mathrm{d}\overline{\boldsymbol{w}}, \tag{6}$$

---

[2]The assumptions are very similar to those in prior work [49]. We list them for completeness in Appendix A.1.

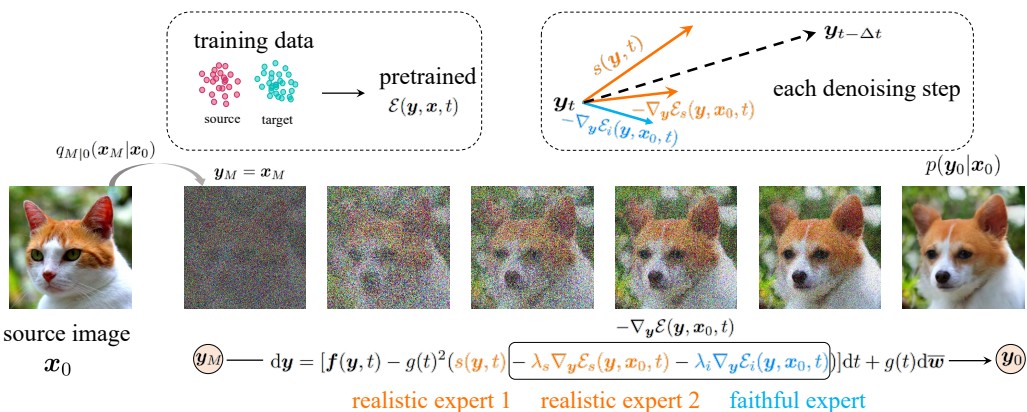

realistic expert 1    realistic expert 2    faithful expert

Figure 2: The overview of our EGSDE. Starting from the noisy source image, we can run the EGSDE for unpaired I2I, which employs an energy function $\mathcal{E}(\boldsymbol{y}, \boldsymbol{x}, t)$ pretrained on both the source and target domains to guide the inference process of a pretrained SDE ($s(\boldsymbol{y}, t)$, realism expert 1). The energy function is decomposed into two terms further, where the realistic expert 2 $\mathcal{E}_s(\boldsymbol{y}, \boldsymbol{x}, t)$ encourages the transferred image to discard domain-specific features and the faithful expert $\mathcal{E}_i(\boldsymbol{y}, \boldsymbol{x}, t)$ aims to preserve the domain-independent ones.

where $\overline{\boldsymbol{w}}$ is a reverse-time standard Wiener process, $\mathrm{d}t$ is an infinitesimal negative timestep, $\boldsymbol{s}(\cdot, \cdot)$ : $\mathbb{R}^D \times \mathbb{R} \to \mathbb{R}^D$ is the score-based model in the pretrained SDE and $\mathcal{E}(\cdot, \cdot, \cdot) : \mathbb{R}^D \times \mathbb{R}^D \times \mathbb{R} \to \mathbb{R}$ is the energy function. The start point $\boldsymbol{y}_M$ is sampled from the perturbation distribution $q_{M|0}(\boldsymbol{y}_M|\boldsymbol{x}_0)$ [32], where $M = 0.5T$ typically. We obtain the transferred images by taking the samples at endpoint $t = 0$ following the SDE in Eq. (6).

Similar to the prior work [7, 32], EGSDE employs an SDE trained solely in the target domain as in Eq. (2), which defines a marginal distribution of the target images and mainly contributes to the realism of the transferred samples. In contrast, the energy function involves the training data across both the source and target domain, making EGSDE distinct from the prior work [7, 32]. Notably, although many other possibilities exist, we carefully design the energy function such that it (approximately) encourages the sample to retain the domain-independent features and discard the domain-specific ones to improve both the faithfulness and realism of the transferred sample. Below, we formally formulate the energy function.

### 3.1 Choice of Energy

In this section, we show how to design the energy function. Intuitively, during the translation, the domain-independent features (pose, color, *etc*. on Cat $\to$ Dog) should be preserved while the domain-specific features (beard, nose, *etc*. on Cat $\to$ Dog) should be changed accordingly. Motivated by this, we decompose the energy function $\mathcal{E}(\boldsymbol{y}, \boldsymbol{x}, t)$ as the sum of two log potential functions [5]:

$$\mathcal{E}(\boldsymbol{y}, \boldsymbol{x}, t) = \lambda_s \mathcal{E}_s(\boldsymbol{y}, \boldsymbol{x}, t) + \lambda_i \mathcal{E}_i(\boldsymbol{y}, \boldsymbol{x}, t) \tag{7}$$

$$= \lambda_s \mathbb{E}_{q_{t|0}(\boldsymbol{x}_t|\boldsymbol{x})} \mathcal{S}_s(\boldsymbol{y}, \boldsymbol{x}_t, t) - \lambda_i \mathbb{E}_{q_{t|0}(\boldsymbol{x}_t|\boldsymbol{x})} \mathcal{S}_i(\boldsymbol{y}, \boldsymbol{x}_t, t), \tag{8}$$

where $\mathcal{E}_i(\cdot, \cdot, \cdot) : \mathbb{R}^D \times \mathbb{R}^D \times \mathbb{R} \to \mathbb{R}$ and $\mathcal{E}_s(\cdot, \cdot, \cdot) : \mathbb{R}^D \times \mathbb{R}^D \times \mathbb{R} \to \mathbb{R}$ are the log potential functions, $\boldsymbol{x}_t$ is the perturbed source image in the forward SDE, $q_{t|0}(\cdot|\cdot)$ is the perturbation kernel from time $0$ to time $t$ in the forward SDE, $\mathcal{S}_s(\cdot, \cdot, \cdot) : \mathbb{R}^D \times \mathbb{R}^D \times \mathbb{R} \to \mathbb{R}$ and $\mathcal{S}_i(\cdot, \cdot, \cdot) : \mathbb{R}^D \times \mathbb{R}^D \times \mathbb{R} \to \mathbb{R}$ are two functions measuring the similarity between the sample and perturbed source image, and $\lambda_s \in \mathbb{R}_{>0}, \lambda_i \in \mathbb{R}_{>0}$ are two weighting hyper-parameters. Note that the expectation w.r.t. $q_{t|0}(\boldsymbol{x}_t|\boldsymbol{x})$ in Eq. (7) guarantees that the energy function changes slowly over the trajectory to satisfy the regularity conditions in Appendix A.1.

To specify $\mathcal{S}_s(\cdot, \cdot, \cdot)$, we introduce a time-dependent domain-specific feature extractor $E_s(\cdot, \cdot)$ : $\mathbb{R}^D \times \mathbb{R} \to \mathbb{R}^{C \times H \times W}$, where $C$ is the channel-wise dimension, $H$ and $W$ are the dimension of height and width. In particular, $E_s(\cdot, \cdot)$ is the all but the last layer of a classifier that is trained on both domains to predict whether an image is from the source domain or the target domain. Intuitively, $E_s(\cdot, \cdot)$ will preserve the domain-specific features and discard the domain-independent features for accurate predictions. Building upon it, $\mathcal{S}_s(\cdot, \cdot, \cdot)$ is defined as the cosine similarity between the

features extracted from the generated sample and the source image as follows:

$$\mathcal{S}_s(\boldsymbol{y}, \boldsymbol{x}_t, t) = \frac{1}{HW} \sum_{h,w} \frac{E_s^{hw}(\boldsymbol{x}_t, t)^\top E_s^{hw}(\boldsymbol{y}, t)}{||E_s^{hw}(\boldsymbol{x}_t, t)||_2 \, ||E_s^{hw}(\boldsymbol{y}, t)||_2}, \tag{9}$$

where $E_s^{hw}(\cdot, \cdot) \in \mathbb{R}^C$ denote the channel-wise feature at spatial position $(h, w)$. Here we employ the cosine similarity since it preserves the spatial information and helps to improve the FID score empirically (see Appendix C.1 for the ablation study). Intuitively, reducing the energy value in Eq. (7) encourages the transferred sample to discard the domain-specific features to improve realism.

To specify $\mathcal{S}_i(\cdot, \cdot, \cdot)$, we introduce a domain-independent feature extractor $E_i(\cdot, \cdot) : \mathbb{R}^D \times \mathbb{R} \to \mathbb{R}^D$, which is a low-pass filter. Intuitively, $E_i(\cdot, \cdot)$ will preserve the overall structures (i.e., domain-independent features) and discard local information like textures (i.e., domain-specific features). Building upon it, $\mathcal{S}_i(\cdot, \cdot, \cdot)$ is defined as the negative squared $L_2$ distance between the features extracted from the generated sample and source image as follows:

$$\mathcal{S}_i(\boldsymbol{y}, \boldsymbol{x}_t, t) = -||E_i(\boldsymbol{y}, t) - E_i(\boldsymbol{x}_t, t)||_2^2. \tag{10}$$

Here, we choose negative squared $L_2$ distance as the similarity metric because it helps to preserve more domain-independent features empirically (see Appendix C.1 for the ablation study). Intuitively, reducing the energy value in Eq. (7) encourages the transferred sample to preserve the domain-independent features to improve faithfulness. In this paper we employ a low-pass filter for its simpleness and effectiveness while we can train more sophisticated $E_i$, e.g., based on disentangled representation learning methods [42, 6, 14, 23, 28], on the data in the two domains.

In our preliminary experiment, alternative to Eq. (7), we consider a simpler energy function that only involves the original source image $\boldsymbol{x}$ as follows:

$$\mathcal{E}(\boldsymbol{y}, \boldsymbol{x}, t) = \lambda_s \mathcal{S}_s(\boldsymbol{y}, \boldsymbol{x}, t) - \lambda_i \mathcal{S}_i(\boldsymbol{y}, \boldsymbol{x}, t), \tag{11}$$

which does not require to take the expectation w.r.t. $\boldsymbol{x}_t$. We found that it did not perform well because it is not reasonable to measure the similarity between the noise-free source image and the transferred sample in a gradual denoising process. See Appendix C.2 for empirical results.

## 3.2 Solving the Energy-guided Reverse-time SDE

Based on the pretrained score-based model $\boldsymbol{s}(\boldsymbol{y}, t)$ and energy function $\mathcal{E}(\boldsymbol{y}, \boldsymbol{x}, t)$, we can solve the proposed energy-guided SDE to generate samples from conditional distribution $p(\boldsymbol{y}_0|\boldsymbol{x}_0)$. There are numerical solvers to approximate trajectories from SDEs. In this paper, we take the Euler-Maruyama solver following [32] for a fair comparison. Given the EGSDE as in Eq. (6) and adopting a step size $h$, the iteration rule from $s$ to $t = s - h$ is:

$$\boldsymbol{y}_t = \boldsymbol{y}_s - [\boldsymbol{f}(\boldsymbol{y}, s) - g(s)^2(\boldsymbol{s}(\boldsymbol{y}_s, s) - \nabla_{\boldsymbol{y}} \mathcal{E}(\boldsymbol{y}_s, \boldsymbol{x}_0, s))]h + g(s)\sqrt{h}\boldsymbol{z}, \quad \boldsymbol{z} \sim \mathcal{N}(\boldsymbol{0}, \boldsymbol{I}). \tag{12}$$

The expectation in $\mathcal{E}(\boldsymbol{y}_s, \boldsymbol{x}_0, s)$ is estimated by the Monte Carlo method of a single sample for efficiency. For brevity, we present the general sampling procedure of our method in Algorithm 1. In experiments, we use the variance preserve energy-guided SDE (VP-EGSDE) [49, 16] and the details are explained in Appendix A.3, where we can modify the noise prediction network to $\tilde{\epsilon}(\boldsymbol{y}, \boldsymbol{x}_0, t) = \epsilon(\boldsymbol{y}, t) + \sqrt{\bar{\beta}_t} \nabla_{\boldsymbol{y}} \mathcal{E}(\boldsymbol{y}, \boldsymbol{x}_0, t)$ and take it into the sampling procedure in DDPM [16]. Following SDEdit [32], we further extend this by repeating the Algorithm 1 $K$ times (see details in Appendix A.2). Further, we explain the connection with classifier guidance[9] in Appendix A.5.

## 3.3 EGSDE as Product of Experts

Inspired by the posterior inference process in diffusion models [46], we present a *product of experts* [15] explanation for the discretized sampling process of EGSDE, which formalizes our motivation in an alternative perspective and provides insights on the role of each component in EGSDE.

We first define a conditional distribution $\tilde{p}(\boldsymbol{y}_t|\boldsymbol{x}_0)$ at time $t$ as a product of experts:

$$\tilde{p}(\boldsymbol{y}_t|\boldsymbol{x}_0) = \frac{p_{r1}(\boldsymbol{y}_t|\boldsymbol{x}_0)p_e(\boldsymbol{y}_t|\boldsymbol{x}_0)}{Z_t}, \tag{13}$$

---

**Algorithm 1** EGSDE for unpaired image-to-image translation

---

**Require:** the source image $\boldsymbol{x}_0$, the initial time $M$, denoising steps $N$, weighting hyper-parameters
$\lambda_s, \lambda_i$, the similarity function $\mathcal{S}_s(\cdot, \cdot, \cdot), \mathcal{S}_i(\cdot, \cdot, \cdot)$, the score function $\boldsymbol{s}(\cdot, \cdot)$
$\quad \boldsymbol{y} \sim q_{M|0}(\boldsymbol{y}|\boldsymbol{x}_0)$ # the start point
$\quad h = \frac{M}{N}$
$\quad$ **for** $i = N$ to 1 **do**
$\quad\quad s \leftarrow ih$
$\quad\quad \boldsymbol{x} \sim q_{s|0}(\boldsymbol{x}|\boldsymbol{x}_0)$ # sample perturbed source image from the perturbation kernel
$\quad\quad \mathcal{E}(\boldsymbol{y}, \boldsymbol{x}, s) \leftarrow \lambda_s \mathcal{S}_s(\boldsymbol{y}, \boldsymbol{x}, s) - \lambda_i \mathcal{S}_i(\boldsymbol{y}, \boldsymbol{x}, s)$ # compute energy with one Monte Carlo
$\quad\quad \boldsymbol{y} \leftarrow \boldsymbol{y} - [\boldsymbol{f}(\boldsymbol{y}, s) - g(s)^2(\boldsymbol{s}(\boldsymbol{y}, s) - \nabla_{\boldsymbol{y}} \mathcal{E}(\boldsymbol{y}, \boldsymbol{x}, s))]h$ # the update rule in Eq. (12)
$\quad\quad \boldsymbol{z} \sim \mathcal{N}(\boldsymbol{0}, \boldsymbol{I})$ if $i > 1$, else $\boldsymbol{z} = \boldsymbol{0}$
$\quad\quad \boldsymbol{y} \leftarrow \boldsymbol{y} + g(s)\sqrt{h}\boldsymbol{z}$
$\quad$ **end for**
$\quad \boldsymbol{y}_0 \leftarrow \boldsymbol{y}$
$\quad$ **return** $\boldsymbol{y}_0$

---

where $Z_t$ is the partition function, $p_e(\boldsymbol{y}_t|\boldsymbol{x}_0) \propto \exp(-\mathcal{E}(\boldsymbol{y}_t, \boldsymbol{x}_0, t))$ and $p_{r1}(\boldsymbol{y}_t|\boldsymbol{x}_0)$ is the marginal distribution at time $t$ defined by SDEdit based on a pretrained SDE on the target domain.

To sample from $\tilde{p}(\boldsymbol{y}_t|\boldsymbol{x}_0)$, we need to construct a transition kernel $\tilde{p}(\boldsymbol{y}_t|\boldsymbol{y}_s)$, where $t = s - h$ and $h$ is small. Following [46], using the desirable equilibrium $\tilde{p}(\boldsymbol{y}_t|\boldsymbol{x}_0) = \int \tilde{p}(\boldsymbol{y}_t|\boldsymbol{y}_s)\tilde{p}(\boldsymbol{y}_s|\boldsymbol{x}_0)d\boldsymbol{y}_s$, we construct the $\tilde{p}(\boldsymbol{y}_t|\boldsymbol{y}_s)$ as follows:

$$\tilde{p}(\boldsymbol{y}_t|\boldsymbol{y}_s) = \frac{p(\boldsymbol{y}_t|\boldsymbol{y}_s)p_e(\boldsymbol{y}_t|\boldsymbol{x}_0)}{\tilde{Z}_t(\boldsymbol{y}_s)}, \tag{14}$$

where $\tilde{Z}_t(\boldsymbol{y}_s)$ is the partition function and $p(\boldsymbol{y}_t|\boldsymbol{y}_s) = \mathcal{N}(\boldsymbol{\mu}(\boldsymbol{y}_s, h), \Sigma(s, h)\boldsymbol{I})$ is the transition kernel of the pretrained SDE in Eq. (4), i.e., $\boldsymbol{\mu}(\boldsymbol{y}_s, h) = \boldsymbol{y}_s - [\boldsymbol{f}(\boldsymbol{y}_s, s) - g(s)^2\boldsymbol{s}(\boldsymbol{y}_s, s)]h$ and $\Sigma(s, h) = g(s)^2 h$. Assuming that $\mathcal{E}(\boldsymbol{y}_t, \boldsymbol{x}_0, t)$ has low curvature relative to $\Sigma(s, h)^{-1}$, it can be approximated using Taylor expansion around $\boldsymbol{\mu}(\boldsymbol{y}_s, h)$ and further we can obtain

$$\tilde{p}(\boldsymbol{y}_t|\boldsymbol{y}_s) \approx \mathcal{N}(\boldsymbol{\mu}(\boldsymbol{y}_s, h) - \Sigma(s, h)\nabla_{\boldsymbol{y}'}\mathcal{E}(\boldsymbol{y}', \boldsymbol{x}_0, t)|_{\boldsymbol{y}'=\boldsymbol{\mu}(\boldsymbol{y}_s, h)}, \Sigma(s, h)\boldsymbol{I}). \tag{15}$$

More details about derivation are available in Appendix A.4. We can observe the transition kernel $\tilde{p}(\boldsymbol{y}_t|\boldsymbol{y}_s)$ in (15) is equal to the discretization of our EGSDE in Eq. (12). Therefore, solving the energy-guided SDE in a discretization manner is approximately equivalent to drawing samples from a product of experts in Eq. (13). Note that $\mathcal{E}(\boldsymbol{y}_t, \boldsymbol{x}_0, t) = \lambda_s \mathcal{E}_s(\boldsymbol{y}_t, \boldsymbol{x}_0, t) + \lambda_i \mathcal{E}_i(\boldsymbol{y}_t, \boldsymbol{x}_0, t)$, the $\tilde{p}(\boldsymbol{y}_t|\boldsymbol{x}_0)$ can be rewritten as:

$$\tilde{p}(\boldsymbol{y}_t|\boldsymbol{x}_0) = \frac{p_{r1}(\boldsymbol{y}_t|\boldsymbol{x}_0)p_{r2}(\boldsymbol{y}_t|\boldsymbol{x}_0)p_f(\boldsymbol{y}_t|\boldsymbol{x}_0)}{Z_t}, \tag{16}$$

where $p_{r2}(\boldsymbol{y}_t|\boldsymbol{x}_0) \propto \exp(-\lambda_s \mathcal{E}_s(\boldsymbol{y}_t, \boldsymbol{x}_0, t)), p_f(\boldsymbol{y}_t|\boldsymbol{x}_0) \propto \exp(-\lambda_i \mathcal{E}_i(\boldsymbol{y}_t, \boldsymbol{x}_0, t))$.

In Eq. (16), by setting $t = 0$, we can explain that the transferred samples approximately follow the distribution defined by the product of three experts, where $p_{r1}(\boldsymbol{y}_t|\boldsymbol{x}_0)$ and $p_{r2}(\boldsymbol{y}_t|\boldsymbol{x}_0)$ are the *realism experts* and $p_f(\boldsymbol{y}_t|\boldsymbol{x}_0)$ is the *faithful expert*, corresponding to the score function $s(\boldsymbol{y}, t)$ and the log potential functions $\mathcal{E}_s(\boldsymbol{y}, \boldsymbol{x}, t)$ and $\mathcal{E}_i(\boldsymbol{y}, \boldsymbol{x}, t)$ respectively. Such a formulation clearly explains the role of each expert in EGSDE and supports our empirical results.

## 4 Related work

Apart from the prior work mentioned before, we discuss other related work including GANs-based methods for unpaired I2I and SBDMs-based methods for image translation.

**GANs-based methods for Unpaired I2I.** Although previous paired image translation methods have also achieved remarkable performances [18, 51, 50, 37, 55, 59], we mainly focus on unpaired image translation in this work. The methods for two-domain unpaired I2I are mainly divided into two classes: two-side and one-side mapping [35, 57]. In the two-side framework [60, 54, 25, 29, 27, 24, 11, 1, 53, 56, 21], the cycle-consistency constraint is the most widely-used strategy such as in

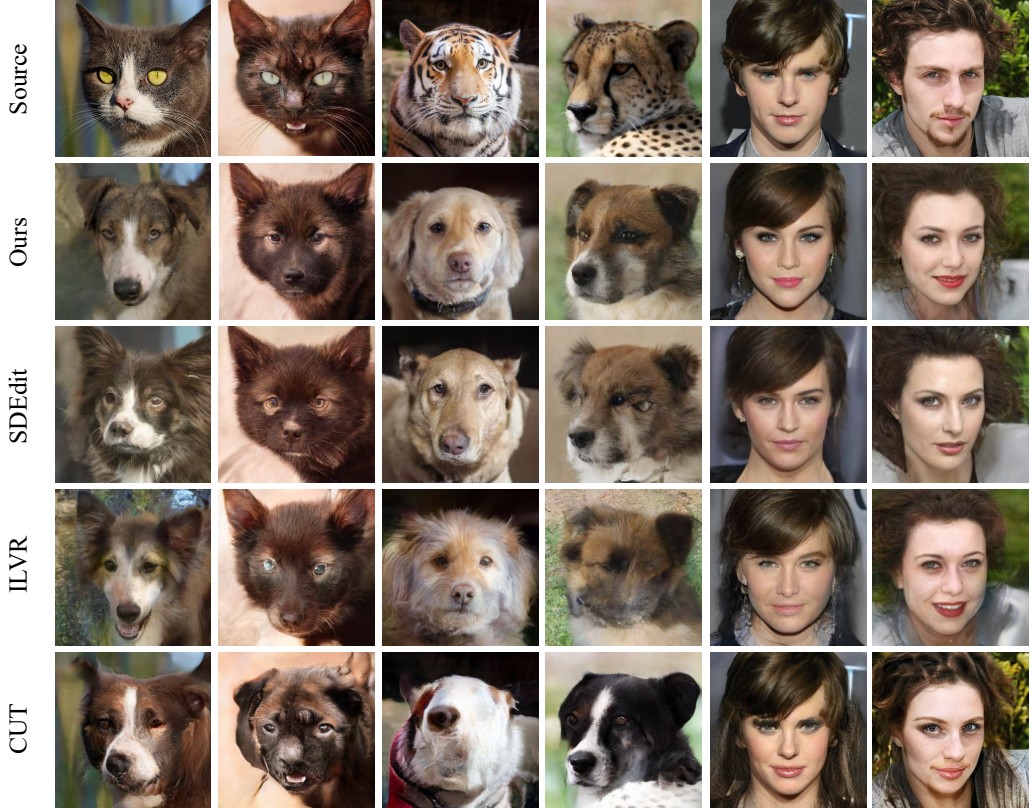

Figure 3: The qualitative comparison on Cat → Dog, Wild → Dog and Male → Female. Our method achieved better visual quality for both *realism* and *faithfulness*. For example, in the forth column, we successfully preserve the domain-independent features (i.e. green ground, pose and yellow color of body) and discard the domain-specific ones (i.e. leopard print).

CycleGAN [60], DualGAN [54] and DiscoGAN [25]. The key idea is that the translated image should be able to be reconstructed by an inverse mapping. More recently, there are numerical studies to improve this such as SCAN [27] and U-GAT-IT [24]. Specifically, U-GAT-IT [24] applies an attention module to let the generator and discriminator focus on more important regions instead of the whole regions through the auxiliary classifier. Since such bijective projection is too restrictive, several studies are devoted to one-side mapping [36, 3, 10, 60, 57, 38, 19]. One representative approach is to design some kind of geometry distance to preserve content [35]. For example, DistanceGAN [3] keeps the distances between images within domains. GCGAN [10] maintains geometry-consistency between input and output. CUT [36] maximizes the mutual information between the input and output using contrastive learning. LSeSim [57] learns spatially-correlative representation to preserve scene structure consistency via self-similarities.

**SBDMs-based methods for Image Translation.** Several studies leveraged SBDMs for image translation due to their powerful generative ability and achieved good results. For example, Diffusion-CLIP [22] fine-tune the score network with CLIP [39] loss, which is applied on text-driven image manipulation, zero-shot image manipulation and multi-attribute transfer successfully. GLIDE [33] and SDG [30] has achieved great performance on text-to-image translation. As for I2I, SR3 [41] and Palette [40] learn a conditional SBDM and outperform state-of-art GANs-based methods on super-resolution, colorization and so on, which needs paired data. For unpaired I2I, UNIT-DDPM [43] learns two SBDMs and two domain translation models using cycle-consistency loss. Compared with it, our method only needs one SBDM on the target domain, which is a kind of one-side mapping. ILVR [7] and SDEdit [32] utilize a SBDM on the target domain and exploited the test source image to refine inference, which ignored the training data in the source domain. Compared with these methods, our method employs an energy function pretrained across both the source and target domains to improve the realism and faithfulness of translated images.

# 5 Experiment

**Datasets.** We validated the EGSDE on following datasets, where all images are resized to $256 \times 256$: (1) CelebA-HQ [20] contains high quality face images and is separated into two domains: male and female. Each category has 1000 testing images. We perform Male→Female on this dataset. (2) AFHQ [8] consists of high-resolution animal face images including three domains: cat, dog and wild, which has relatively large variations. Each domain has 500 testing images. We perform Cat→Dog and Wild→Dog on this dataset. We also perform *multi-domain translation* on AFHQ dataset and the experimental results are reported in Appendix D.

**Implementation.** The time-dependent domain-specific extractor $E_s(\boldsymbol{x}, t)$ is trained based on the backbone in [9]. The resize function including downsampling and upsampling operation is used as low-pass filter and is implemented by [45]. For generation process, by default, the weight parameter $\lambda_s$, $\lambda_i$ is set 500 and 2 respectively. The initial time $M$ and denoising steps $N$ is set $0.5T$ and 500 by default. More details about implementation are available in Appendix B.

**Evaluation Metrics.** We evaluate translated images from two aspects: realism and faithfulness. For realism, we report the widely-used Frechet Inception Score (FID) [13] between translated images and the target dataset. To quantify faithfulness, we report the $L_2$ distance, PSNR and SSIM [52] between each input-output pair. To quantify both faithfulness and realism, we leverage Amazon Mechanical Turk(AMT) human evaluation to perform pairwise comparisons between the baselines and EGSDE. More details is available in Appendix B.6.

## 5.1 Two-Domain Unpaired Image Translation

In this section, we compare EGSDE with the following state-of-the-art I2I methods in three tasks: SBDMs-based methods including ILVR [7] and SDEdit [32], and GANs-based methods including CUT [36], which are reproduced using public code. On the most popular benchmark Cat → Dog, we also report the performance of other state-of-the-art GANs-methods , where StarGAN v2 [8] is evaluated by the provided public checkpoint and the others are public results from CUT[36] and ITTR [58]. We provide more details about reproductions in Appendix B.7.

The quantitative comparisons and qualitative results are shown in Table 1 and Figure 3. We can derive several observations. *First*, our method outperforms the SBDMs-based methods significantly in almost all realism and faithfulness metrics, suggesting the effectiveness of employing energy function pretrained on both domains to guide the generation process. Especially, compared with the most direct competitor, i.e., SDEdit, with a lower $L_2$ distance at the same time, EGSDE improves the FID score by 8.35, 8.76 and 7.5 on Cat → Dog, Wild → Dog and Male → Female respectively. *Second*, EGSDE[†] outperforms the current state-of-art GANs-based methods by a large margin on the challenging AFHQ dataset. For example, compared with CUT [36], we achieve an improvement of FID score with 25.17 and 42.51 on the Cat → Dog and Wild → Dog tasks respectively. In addition, the human evaluation shows that EGSDE are preferred compared to all baselines ($> 50\%$). The qualitative results in Figure 3 agree with quantitative comparisons in Table 1, where our method achieved the results with the best visual quality for both realism and faithfulness. We show more qualitative results and select some failure cases in Appendix C.6.

## 5.2 Ablation Studies

**The function of each expert.** We validate the function of *realistic expert* $\mathcal{E}_s(\boldsymbol{y}, \boldsymbol{x}, t)$ and *faithful expert* $\mathcal{E}_i(\boldsymbol{y}, \boldsymbol{x}, t)$ by changing the weighting hyper-parameter $\lambda_s$ and $\lambda_i$. As shown in Table 3 and Figure 1, larger $\lambda_s$ results in more realistic images and larger $\lambda_i$ results in more faithful images. More results is available in Appendix C.5.

**The choice of initial time $M$.** We explore the effect of the initial time $M$ of EGSDE. As shown in Figure 4, the larger $M$ results in more realistic and less faithful image. More results is available in Appendix C.3.

**Repeating $K$ Times.** Following SDEdit [32], we show the results of repeating the Algorithm 1 $K$ times. The quantitative and qualitative results are depicted in Table 2 and Figure 4. The experimental results show the EGSDE outperforms SDEdit in each $K$ step in all metrics. With the increase of $K$, the SDEdit generates more realism images but the faithful metrics decrease sharply, because it only

Table 1: Quantitative comparison. ILVR [7], SDEdit [32] and CUT [36] are reproduced using public code. StarGAN v2 [8] is evaluated by the provided public checkpoint and the other methods marked by * are public results from CUT[36] and ITTR [58]. All SBDMs-based methods and StarGAN v2 are repeated 5 times to eliminate randomness. CUT is conducted once since it learns a deterministic mapping. AMT show the preference rate of EGSDE against baselines via human evaluation. The EGSDE use the default-parameters ($\lambda_s = 500, \lambda_i = 2, M = 0.5T$) and EGSDE$^{\dagger}$ use the parameters with $\lambda_s = 700, \lambda_i = 0.5, M = 0.6T$.

| Model | FID $\downarrow$ | L2 $\downarrow$ | PSNR $\uparrow$ | SSIM $\uparrow$ | AMT $\uparrow$ |
|---|---|---|---|---|---|
| | | Cat $\rightarrow$ Dog | | | |
| CycleGAN* [60] | 85.9 | - | - | - | - |
| MUNIT* [17] | 104.4 | - | - | - | - |
| DRIT* [26] | 123.4 | - | - | - | - |
| Distance* [3] | 155.3 | - | - | - | - |
| SelfDistance* [3] | 144.4 | - | - | - | - |
| GCGAN* [10] | 96.6 | - | - | - | - |
| LSeSim* [57] | 72.8 | - | - | - | - |
| ITTR (CUT)* [58] | 68.6 | - | - | - | - |
| StarGAN v2 [8] | 54.88 ± 1.01 | 133.65 ± 1.54 | 10.63 ± 0.10 | 0.27 ± 0.003 | - |
| CUT* [36] | 76.21 | 59.78 | 17.48 | **0.601** | 79.6% |
| ILVR [7] | 74.37 ± 1.55 | 56.95 ± 0.14 | 17.77 ± 0.02 | 0.363 ± 0.001 | 75.4% |
| SDEdit [32] | 74.17 ± 1.01 | 47.88 ± 0.06 | 19.19 ± 0.01 | **0.423 ± 0.001** | 65.2% |
| EGSDE | **65.82 ± 0.77** | **47.22 ± 0.08** | **19.31 ± 0.02** | 0.415 ± 0.001 | - |
| EGSDE$^{\dagger}$ | **51.04 ± 0.37** | 62.06 ± 0.10 | 17.17 ± 0.02 | 0.361 ± 0.001 | - |
| | | Wild $\rightarrow$ Dog | | | |
| CUT [36] | 92.94 | 62.21 | 17.2 | **0.592** | 82.4% |
| ILVR [7] | 75.33 ± 1.22 | 63.40 ± 0.15 | 16.85 ± 0.02 | 0.287 ± 0.001 | 73.4% |
| SDEdit [32] | 68.51 ± 0.65 | 55.36 ± 0.05 | 17.98 ± 0.01 | 0.343 ± 0.001 | 57.2% |
| EGSDE | **59.75 ± 0.62** | **54.34 ± 0.08** | **18.14 ± 0.01** | 0.343 ± 0.001 | - |
| EGSDE$^{\dagger}$ | **50.43± 0.52** | 66.52± 0.09 | 16.40± 0.01 | 0.300± 0.001 | - |
| | | Male $\rightarrow$ Female | | | |
| CUT [36] | **31.94** | 46.61 | 19.87 | **0.74** | 58.6% |
| ILVR [7] | 46.12 ± 0.33 | 52.17 ± 0.10 | 18.59 ± 0.02 | 0.510 ± 0.001 | 88.2% |
| SDEdit [32] | 49.43 ± 0.47 | 43.70 ± 0.03 | 20.03 ± 0.01 | 0.572 ± 0.000 | 74.4% |
| EGSDE | **41.93 ± 0.11** | **42.04 ± 0.03** | **20.35 ± 0.01** | **0.574 ± 0.000** | - |
| EGSDE$^{\dagger}$ | **30.61 ± 0.19** | 53.44 ± 0.09 | 18.32 ± 0.02 | 0.510 ± 0.001 | - |

utilizes the source image at the initial time $M$. As shown in Figure 4, when $K$=3, SDEdit discard the domain-independent information of the source image (i.e., color and background) while our method still preserves them without harming realism.

## 6  Conclusions and Discussions

In this paper, we propose energy-guided stochastic differential equations (EGSDE) for realistic and faithful unpaired I2I, which employs an energy function pretrained on both domains to guide the generation process of a pretrained SDE. Building upon two feature extractors, we carefully design the energy function to preserve the domain-independent features and discard domain-specific ones of the source image. We demonstrate the EGSDE by outperforming state-of-art I2I methods on three widely-adopted unpaired I2I tasks.

One limitation of this paper is we employ a low-pass filter as the domain-independent feature extractor for its simpleness and effectiveness while we can train more sophisticated extractor, e.g. based on disentangled representation learning methods [42, 6, 14, 23, 28], on the data in the two domains. We

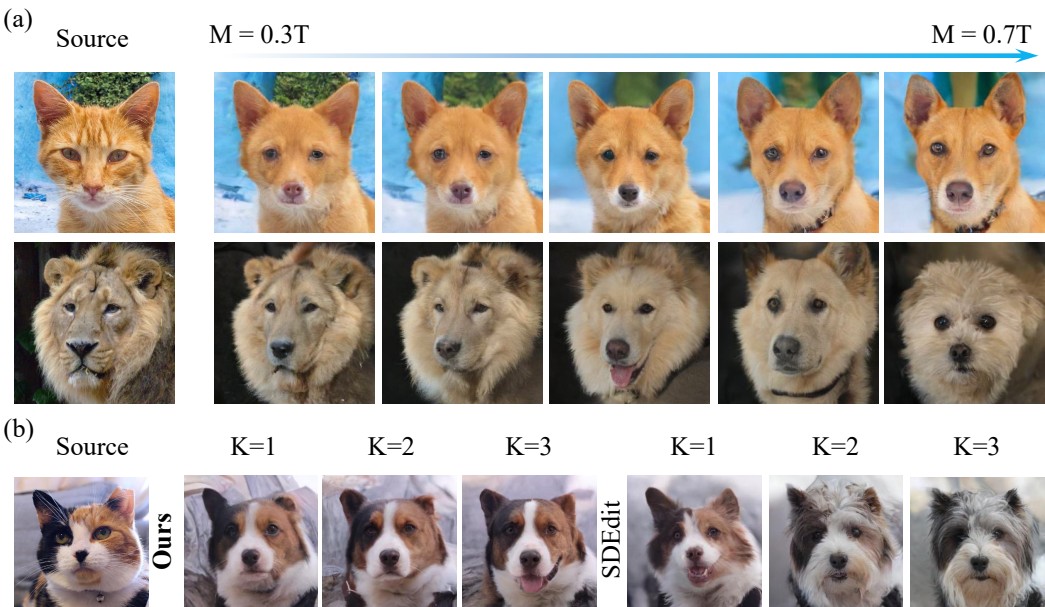

Figure 4: (a) The results of different initial time $M$. The larger $M$ results in more realistic and less faithful images. (b) The results of repeating the Algorithm 1 $K$ times. With the increase of K, SDEdit [32] tend to discard the domain-independent information of the source image (e.g., color and background) while our method still preserve them without harming realism.

Table 2: Comparison with SDEdit [32] under different $K$ times on Male → Female. The results on other tasks are reported in Appendix C.4.

| Methods | K | FID↓ | L2↓ | PSNR↑ | SSIM↑ |
|---|---|---|---|---|---|
| SDEdit [32] | 1 | 49.95 | 43.71 | 20.03 | 0.572 |
| EGSDE | | **42.17** | **42.07** | **20.35** | **0.573** |
| SDEdit [32] | 2 | 46.26 | 50.70 | 18.77 | 0.542 |
| EGSDE | | **38.68** | **47.10** | **19.40** | **0.548** |
| SDEdit [32] | 3 | 45.19 | 55.03 | 18.08 | 0.527 |
| EGSDE | | **37.55** | **49.63** | **18.96** | **0.536** |

Table 3: The results of different $\lambda_s$ and $\lambda_i$ on Wild → Dog. $\lambda_s = \lambda_i = 0$ corresponds to SDEdit [32].

| $\lambda_s, \lambda_i$ | FID↓ | L2↓ | PSNR↑ | SSIM↑ |
|---|---|---|---|---|
| $\lambda_s = 0, \lambda_i = 0$ | 67.87 | 55.39 | 17.97 | 0.344 |
| $\lambda_s = 100, \lambda_i = 0$ | 60.80 | 56.19 | 17.85 | 0.341 |
| $\lambda_s = 500, \lambda_i = 0$ | 53.72 | 58.65 | 17.47 | 0.335 |
| $\lambda_s = 800, \lambda_i = 0$ | 53.01 | 60.02 | 17.27 | 0.331 |
| $\lambda_s = 0, \lambda_i = 0.5$ | 68.31 | 53.23 | 18.32 | 0.347 |
| $\lambda_s = 0, \lambda_i = 2$ | 71.10 | 51.99 | 18.52 | 0.349 |
| $\lambda_s = 0, \lambda_i = 5$ | 72.70 | 51.44 | 18.61 | 0.351 |

leave this issue in future work. In addition, we must take care to exploit the method to avoid the potential negative social impact (i.e., generating fake images to mislead people).

## Acknowledgement

We thank Cheng Lu, Yuhao Zhou, Haoyu Liang and Shuyu Cheng for helpful discussions about the method and its limitations. This work was supported by the National Key Research and Development Program of China (2020AAA0106302); NSF of China Projects (Nos. 62061136001, 61620106010, 62076145, U19B2034, U1811461, U19A2081, 6197222); Beijing NSF Project (No. JQ19016); Beijing Outstanding Young Scientist Program NO. BJJWZYJH012019100020098; a grant from Tsinghua Institute for Guo Qiang; the High Performance Computing Center, Tsinghua University; the Fundamental Research Funds for the Central Universities, and the Research Funds of Renmin University of China (22XNKJ13). Part of the computing resources supporting this work, totaled 500 A100 GPU hours, were provided by High-Flyer AI. (Hangzhou High-Flyer AI Fundamental Research Co., Ltd.). J.Z was also supported by the XPlorer Prize.

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
