# A Details about EGSDE

## A.1 Assumptions about EGSDE

**Notations.** $\boldsymbol{f}(\cdot, \cdot) : \mathbb{R}^D \times \mathbb{R} \to \mathbb{R}^D$ is the drift coefficient. $g(\cdot) : \mathbb{R} \to \mathbb{R}$ is the diffusion coefficient. $\boldsymbol{s}(\cdot, \cdot) : \mathbb{R}^D \times \mathbb{R} \to \mathbb{R}^D$ is the score-based model. $\mathcal{E}(\cdot, \cdot, \cdot) : \mathbb{R}^D \times \mathbb{R}^D \times \mathbb{R} \to \mathbb{R}$ is the energy function. $\boldsymbol{x}_0$ is the given source image.

**Assumptions.** EGSDE defines a valid conditional distribution $p(\boldsymbol{y}_0|\boldsymbol{x}_0)$ under following assumptions:

(1) $\exists C > 0, \forall t \in \mathbb{R}, \forall \boldsymbol{x}, \boldsymbol{y} \in \mathbb{R}^D : ||f(\boldsymbol{x}, t) - f(\boldsymbol{y}, t)||_2 \leq C||\boldsymbol{x} - \boldsymbol{y}||_2$.

(2) $\exists C > 0, \forall t, s \in \mathbb{R}, \forall \boldsymbol{y} \in \mathbb{R}^D : ||f(\boldsymbol{y}, t) - f(\boldsymbol{y}, s)||_2 \leq C|t - s|$.

(3) $\exists C > 0, \forall t \in \mathbb{R}, \forall \boldsymbol{x}, \boldsymbol{y} \in \mathbb{R}^D : ||\boldsymbol{s}(\boldsymbol{x}, t) - \boldsymbol{s}(\boldsymbol{y}, t)||_2 \leq C||\boldsymbol{x} - \boldsymbol{y}||_2$.

(4) $\exists C > 0, \forall t, s \in \mathbb{R}, \forall \boldsymbol{y} \in \mathbb{R}^D : ||\boldsymbol{s}(\boldsymbol{y}, t) - \boldsymbol{s}(\boldsymbol{y}, s)||_2 \leq C|t - s|$.

(5) $\exists C > 0, \forall t \in \mathbb{R}, \forall \boldsymbol{x}, \boldsymbol{y} \in \mathbb{R}^D : ||\nabla_{\boldsymbol{x}}\mathcal{E}(\boldsymbol{x}, \boldsymbol{x}_0, t) - \nabla_{\boldsymbol{y}}\mathcal{E}(\boldsymbol{y}, \boldsymbol{x}_0, t)||_2 \leq C||\boldsymbol{x} - \boldsymbol{y}||_2$.

(6) $\exists C > 0, \forall t, s \in \mathbb{R}, \forall \boldsymbol{y} \in \mathbb{R}^D : ||\nabla_{\boldsymbol{y}}\mathcal{E}(\boldsymbol{y}, \boldsymbol{x}_0, t) - \nabla_{\boldsymbol{y}}\mathcal{E}(\boldsymbol{y}, \boldsymbol{x}_0, s)||_2 \leq C|t - s|$.

(7) $\exists C > 0, \forall t, s \in \mathbb{R} : |g(t) - g(s)| \leq C|t - s|$.

## A.2 An Extention of EGSDE

Following SDEdit [9], we further extend the original EGSDE by repeating it $K$ times. The general sampling procedure is summarized in Algorithm 1.

---
**Algorithm 1** An extention of EGSDE for unpaired image-to-image translation

---
**Require:** the source image $\boldsymbol{x}_0$, the initial time $M$, denoising steps $N$, weighting hyper-parameters $\lambda_s, \lambda_i$, the similarity function $\mathcal{S}_s(\cdot, \cdot, \cdot), \mathcal{S}_i(\cdot, \cdot, \cdot)$, the score function $\boldsymbol{s}(\cdot, \cdot)$, repeating times $K$

  $h = \frac{M}{N}$
  $\boldsymbol{y}_0 \leftarrow \boldsymbol{x}_0$
  **for** $k = 1$ to $K$ **do**
    $\boldsymbol{y} \sim q_{M|0}(\boldsymbol{y}|\boldsymbol{y}_0)$ # the start point
    **for** $i = N$ to 1 **do**
      $s \leftarrow ih$
      $\boldsymbol{x} \sim q_{s|0}(\boldsymbol{x}|\boldsymbol{x}_0)$ # sample perturbed source image from the perturbation kernel
      $\mathcal{E}(\boldsymbol{y}, \boldsymbol{x}, s) \leftarrow \lambda_s \mathcal{S}_s(\boldsymbol{y}, \boldsymbol{x}, s) - \lambda_i \mathcal{S}_i(\boldsymbol{y}, \boldsymbol{x}, s)$ # compute energy with one Monte Carlo
      $\boldsymbol{y} \leftarrow \boldsymbol{y} - [\boldsymbol{f}(\boldsymbol{y}, s) - g(s)^2(\boldsymbol{s}(\boldsymbol{y}, s) - \nabla_{\boldsymbol{y}}\mathcal{E}(\boldsymbol{y}, \boldsymbol{x}, s))]h$
      $\boldsymbol{z} \sim \mathcal{N}(\boldsymbol{0}, \boldsymbol{I})$ if $i > 1$, else $\boldsymbol{z} = \boldsymbol{0}$
      $\boldsymbol{y} \leftarrow \boldsymbol{y} + g(s)\sqrt{h}\boldsymbol{z}$
    **end for**
    $\boldsymbol{y}_0 \leftarrow \boldsymbol{y}$
  **end for**
  $\boldsymbol{y}_0 \leftarrow \boldsymbol{y}$
  **return** $\boldsymbol{y}_0$

---

## A.3 Variance Preserve Energy-guided SDE (VP-EGSDE)

In this section, we show a specific EGSDE: variance preserve energy-guided SDE (VP-EGSDE) [12, 7], which is conducted in our experiments. The VP-EGSDE is defined as follows:

$$\mathrm{d}\boldsymbol{y} = [-\frac{1}{2}\beta(t)\boldsymbol{y} - \beta(t)(\boldsymbol{s}(\boldsymbol{y}, t) - \nabla_{\boldsymbol{y}}\mathcal{E}(\boldsymbol{y}, \boldsymbol{x}_0, t))]\mathrm{d}t + \sqrt{\beta(t)}\mathrm{d}\overline{\boldsymbol{w}}, \tag{1}$$

where $\boldsymbol{x}_0$ is the given source image, $\beta(\cdot) : \mathbb{R} \to \mathbb{R}$ is a positive function, $\overline{\boldsymbol{w}}$ is a reverse-time standard Wiener process, $\mathrm{d}t$ is an infinitesimal negative timestep, $\boldsymbol{s}(\cdot, \cdot) : \mathbb{R}^D \times \mathbb{R} \to \mathbb{R}^D$ is the score-based model in the pretrained SDE and $\mathcal{E}(\cdot, \cdot, \cdot) : \mathbb{R}^D \times \mathbb{R}^D \times \mathbb{R} \to \mathbb{R}$ is the energy function. The perturbation kernel $q_{t|0}(\boldsymbol{y}_t|\boldsymbol{y}_0)$ is $\mathcal{N}(\boldsymbol{y}_0 e^{-\frac{1}{2}\int_0^t \beta(s)\mathrm{d}s}, (1 - e^{-\int_0^t \beta(s)\mathrm{d}s})\boldsymbol{I})$ and $\beta(t) =$

$\beta_{min} + t(\beta_{max} - \beta_{min})$ in practice. Following [9, 7], we use $\beta_{min} = 0.1, \beta_{max} = 20$. The iteration rule from $s$ to $t = s - h$ of VP-EGSDE in Eq. (1) is:

$$\boldsymbol{y}_t = \frac{1}{\sqrt{1 - \beta(s)h}}(\boldsymbol{y}_s + \beta(s)h(\boldsymbol{s}(\boldsymbol{y}_s, s) - \nabla_{\boldsymbol{y}}\mathcal{E}(\boldsymbol{y}_s, \boldsymbol{x}_0, s)) + \sqrt{\beta(s)h}\boldsymbol{z}, \quad \boldsymbol{z} \sim \mathcal{N}(\boldsymbol{0}, \boldsymbol{I}), \quad (2)$$

where $h$ is a small step size. [12] showed the iteration rule in Eq. (2) is equivalent to that using Euler-Maruyama solver when $h$ is small in Appendix E. In other words, the score network is modified to $\tilde{s}(\boldsymbol{y}, \boldsymbol{x}_0, t) = \boldsymbol{s}(\boldsymbol{y}, t) - \nabla_{\boldsymbol{y}}\mathcal{E}(\boldsymbol{y}, \boldsymbol{x}_0, t)$ in EGSDE . Accordingly, we can modify the noise prediction network to $\tilde{\epsilon}(\boldsymbol{y}, \boldsymbol{x}_0, t) = \epsilon(\boldsymbol{y}, t) + \sqrt{\bar{\beta}_t}\nabla_{\boldsymbol{y}}\mathcal{E}(\boldsymbol{y}, \boldsymbol{x}_0, t)$ and take it into the sampling procedure in DDPM [7].

---

**Algorithm 2** VP-EGSDE for unpaired image-to-image translation

---

**Require:** the source image $\boldsymbol{x}_0$, the initial time $M$, denoising steps $N$, weighting hyper-parameters $\lambda_s, \lambda_i$, the similarity function $\mathcal{S}_s(\cdot, \cdot, \cdot), \mathcal{S}_i(\cdot, \cdot, \cdot)$, the score function $\boldsymbol{s}(\cdot, \cdot)$
  $\boldsymbol{y} \sim q_{M|0}(\boldsymbol{y}|\boldsymbol{x}_0)$ # the start point
  $h = \frac{M}{N}$
  **for** $i = N$ to 1 **do**
    $s \leftarrow ih$
    $\boldsymbol{x} \sim q_{s|0}(\boldsymbol{x}|\boldsymbol{x}_0)$ # sample perturbed source image from the perturbation kernel
    $\mathcal{E}(\boldsymbol{y}, \boldsymbol{x}, s) \leftarrow \lambda_s \mathcal{S}_s(\boldsymbol{y}, \boldsymbol{x}, s) - \lambda_i \mathcal{S}_i(\boldsymbol{y}, \boldsymbol{x}, s)$ # compute energy with one Monte Carlo
    $\boldsymbol{y} \leftarrow \frac{1}{\sqrt{1-\beta(s)h}}(\boldsymbol{y} + \beta(s)h(\boldsymbol{s}(\boldsymbol{y}_s, s) - \nabla_{\boldsymbol{y}}\mathcal{E}(\boldsymbol{y}_s, \boldsymbol{x}, s))$ # the update rule in Eq. (2)
    $\boldsymbol{z} \sim \mathcal{N}(\boldsymbol{0}, \boldsymbol{I})$ if $i > 1$, else $\boldsymbol{z} = \boldsymbol{0}$
    $\boldsymbol{y} \leftarrow \boldsymbol{y} + \sqrt{\beta(s)h}\boldsymbol{z}$
  **end for**
  $\boldsymbol{y}_0 \leftarrow \boldsymbol{y}$
  **return** $\boldsymbol{y}_0$

---

### A.4 EGSDE as Product of Experts

In this section, we provide more details about the *product of experts* [6] explanation for the discretized sampling process of EGSDE. Recall that we construct the $\tilde{p}(\boldsymbol{y}_t|\boldsymbol{y}_s)$ as follows:

$$\tilde{p}(\boldsymbol{y}_t|\boldsymbol{y}_s) = \frac{p(\boldsymbol{y}_t|\boldsymbol{y}_s)p_e(\boldsymbol{y}_t|\boldsymbol{x}_0)}{\tilde{Z}_t(\boldsymbol{y}_s)}, \quad (3)$$

where $\tilde{Z}_t(\boldsymbol{y}_s)$ is the partition function, $p(\boldsymbol{y}_t|\boldsymbol{y}_s) = \mathcal{N}(\boldsymbol{\mu}(\boldsymbol{y}_s, h), \Sigma(s, h)\boldsymbol{I})$ is the transition kernel of the pretrained SDE, i.e., $\boldsymbol{\mu}(\boldsymbol{y}_s, h) = \boldsymbol{y}_s - [\boldsymbol{f}(\boldsymbol{y}_s, s) - g(s)^2\boldsymbol{s}(\boldsymbol{y}_s, s)]h$ and $\Sigma(s, h) = g(s)^2h$. For brevity, we denote $\boldsymbol{\mu} = \boldsymbol{\mu}(\boldsymbol{y}_s, h), \Sigma = \Sigma(s, h)$. Assuming that $\mathcal{E}(\boldsymbol{y}_t, \boldsymbol{x}_0, t)$ has low curvature relative to $\Sigma^{-1}$, then we can use Taylor expansion around $\boldsymbol{\mu}$ to approximate it:

$$\mathcal{E}(\boldsymbol{y}_t, \boldsymbol{x}_0, t) \approx \mathcal{E}(\boldsymbol{\mu}, \boldsymbol{x}_0, t) + (\boldsymbol{y}_t - \boldsymbol{\mu})^{\top}\boldsymbol{g}, \quad (4)$$

where $\boldsymbol{g} = \nabla\boldsymbol{y}'\mathcal{E}(\boldsymbol{y}', \boldsymbol{x}_0, t)|_{\boldsymbol{y}'=\boldsymbol{\mu}}$. Taking it into Eq. (3), we can get:

$$\log\tilde{p}(\boldsymbol{y}_t|\boldsymbol{y}_s) \approx -\frac{1}{2}(\boldsymbol{y}_t - \boldsymbol{\mu})^{\top}\Sigma^{-1}(\boldsymbol{y}_t - \boldsymbol{\mu}) - (\boldsymbol{y}_t - \boldsymbol{\mu})^{\top}\boldsymbol{g} + constant \quad (5)$$

$$= -\frac{1}{2}\boldsymbol{y}_t^{\top}\Sigma^{-1}\boldsymbol{y}_t + \frac{1}{2}\boldsymbol{y}_t^{\top}\Sigma^{-1}\boldsymbol{\mu} + \frac{1}{2}\boldsymbol{\mu}^{\top}\Sigma^{-1}\boldsymbol{y}_t \quad (6)$$

$$- \frac{1}{2}\boldsymbol{y}^{\top}\Sigma^{-1}\Sigma\boldsymbol{g} - \frac{1}{2}\boldsymbol{g}^{\top}\Sigma\Sigma^{-1}\boldsymbol{y} + constant \quad (7)$$

$$= -\frac{1}{2}(\boldsymbol{y}_t - \boldsymbol{\mu} + \Sigma\boldsymbol{g})^{\top}\Sigma^{-1}(\boldsymbol{y}_t - \boldsymbol{\mu} + \Sigma\boldsymbol{g}) + constant. \quad (8)$$

Therefore,

$$\tilde{p}(\boldsymbol{y}_t|\boldsymbol{y}_s) \approx \mathcal{N}(\boldsymbol{\mu} - \Sigma\boldsymbol{g}, \Sigma\boldsymbol{I}) \quad (9)$$

$$= \mathcal{N}(\boldsymbol{\mu} - \Sigma\nabla_{\boldsymbol{y}'}\mathcal{E}(\boldsymbol{y}', \boldsymbol{x}_0, t)|_{\boldsymbol{y}'=\boldsymbol{\mu}}, \Sigma\boldsymbol{I}). \quad (10)$$

Therefore, solving the EGSDE in a discretization manner is approximately equivalent to drawing samples from a product of experts.

| Source | Ours | Source | Ours | Source | Ours |
|--------|------|--------|------|--------|------|

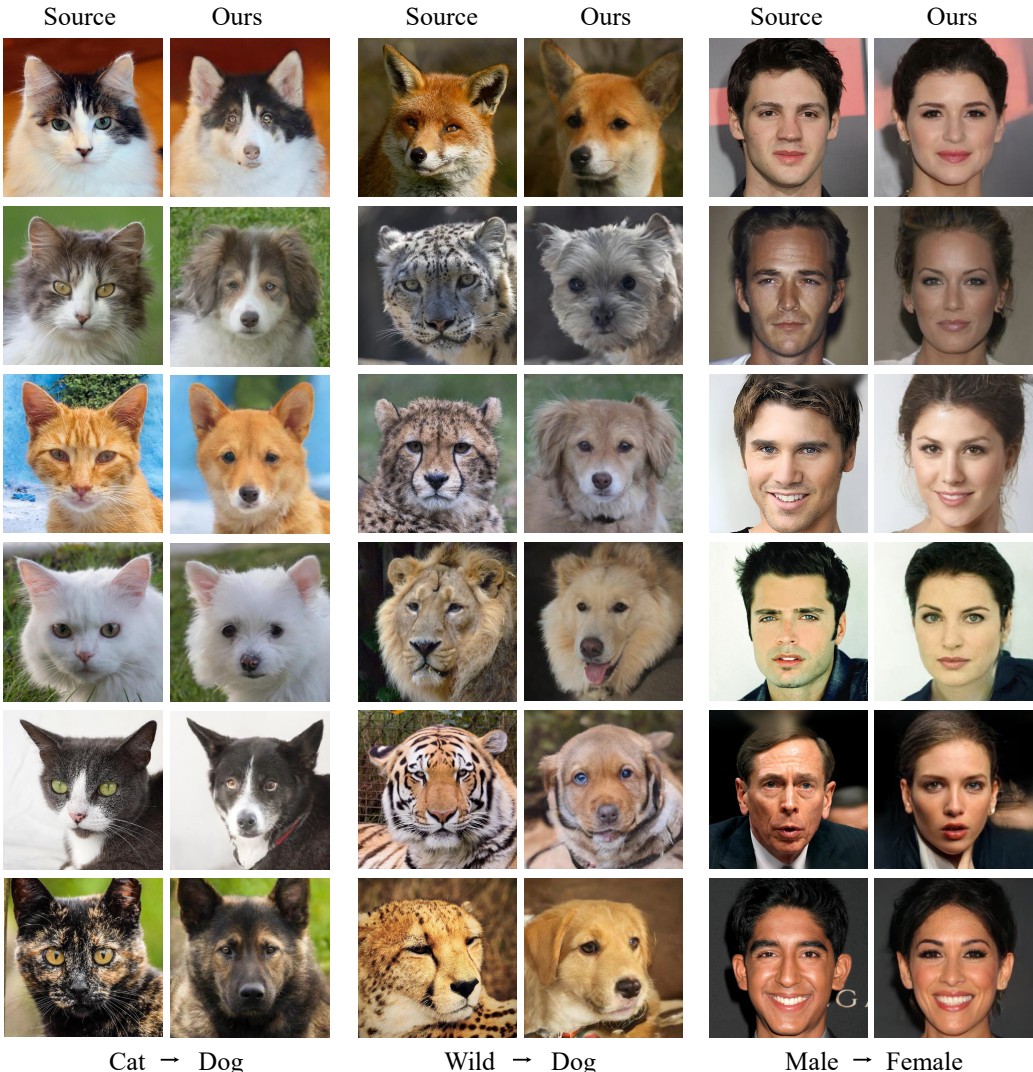

$$\text{Cat} \rightarrow \text{Dog} \qquad \text{Wild} \rightarrow \text{Dog} \qquad \text{Male} \rightarrow \text{Female}$$

Figure 1: More qualitative results on three unpaired I2I tasks.

## A.5 The Connection with Classifier Guidance

In this section, we show the classifier guidance[4] can be regarded as a special design of energy function and provide an alternative explanation of the classifier guidance as a product of experts.

Recall that the EGSDE, which leverages an energy function to guide the inference process of a pretrained SDE, is defined as follows:

$$\mathrm{d}\boldsymbol{x} = [\boldsymbol{f}(\boldsymbol{x},t) - g(t)^2(\boldsymbol{s}(\boldsymbol{x},t) - \nabla_{\boldsymbol{x}}\mathcal{E}(\boldsymbol{x},c,t))]\mathrm{d}t + g(t)\mathrm{d}\overline{\boldsymbol{w}}, \tag{11}$$

which defines a distribution $p(\boldsymbol{x}_0|c)$ conditioned on c. Let $\mathcal{E}(\boldsymbol{x},c,t) \propto -\log p_t(c|\boldsymbol{x})^\lambda$, where $p_t(c|\boldsymbol{x})$ is a time-dependent classifier and $c$ is the class label, the EGSDE can be rewritten as:

$$\mathrm{d}\boldsymbol{x} = [\boldsymbol{f}(\boldsymbol{x},t) - g(t)^2(\boldsymbol{s}(\boldsymbol{x},t) + \lambda\nabla_{\boldsymbol{x}}\log p_t(c|\boldsymbol{x}))]\mathrm{d}t + g(t)\mathrm{d}\overline{\boldsymbol{w}}. \tag{12}$$

Sovling variance preserve energy-guided SDE (VP-EGSDE) in Eq. 12 with Euler-Maruyama solver is equal to the classifier guidance in [4, 12].

Assuming $\int e^{-\mathcal{E}(\boldsymbol{x},c,t)}d\boldsymbol{x} < \infty$, we can define a conditional distribution $q_t(\boldsymbol{x}|c)$ at time $t$ as follows:

$$q_t(\boldsymbol{x}|c) = \frac{e^{-\mathcal{E}(\boldsymbol{x},c,t)}}{\int e^{-\mathcal{E}(\boldsymbol{x},c,t)}d\boldsymbol{x}} \tag{13}$$

Table 1: The used codes and license.

| URL | citations | License |
|---|---|---|
| https://github.com/openai/guided-diffusion | [4] | MIT License |
| https://github.com/taesungp/contrastive-unpaired-translation | [10] | BSD License |
| https://github.com/jychoi118/ilvr_adm | [2] | MIT License |
| https://github.com/ermongroup/SDEdit | [9] | MIT License |
| https://github.com/mseitzer/pytorch-fid | [5] | Apache V2.0 License |

According to the analysis in section **??**, solving the EGSDE in Eq. 12 is approximately equivalent to drawing samples from a product of experts as follows:

$$\tilde{p}_t(\boldsymbol{x}|c) = \frac{p_t(\boldsymbol{x})q_t(\boldsymbol{x}|c)}{Z_t}, \tag{14}$$

where $p_t(\boldsymbol{x})$ is the marginal distribution at time $t$ defined by a pretrained SDE. Therefore, the generated samples in classifier guidance approximately follow the distribution:

$$\tilde{p}_0(\boldsymbol{x}|c) = \frac{p_0(\boldsymbol{x})q_0(\boldsymbol{x}|c)}{Z_0}. \tag{15}$$

Similarly, combining a conditional socre-based model and a classifier in [4] is approximately equivalent to drawing samples from a product of experts as follows:

$$\tilde{p}_0(\boldsymbol{x}|c) = \frac{p_0(\boldsymbol{x}|c)q_0(\boldsymbol{x}|c)}{Z_0}, \tag{16}$$

where $p_0(\boldsymbol{x}|c)$ is the marginal distribution at time $t$ defined by a pretrained SDE.

## B  Implementation Details

### B.1  Datasets

To validate our method, we conduct experiments on the following datasets:

(1) CelebA-HQ [8] contains high quality face images and is separated into two domains: male and female. For training data, it contains 10057 male images and 17943 female images. Each category has 1000 testing images. Male→Female task was conducted on this dataset.

(2) AFHQ [3] consists of high-resolution animal face images including three domains: cat, dog and wild, which has relatively large variations. For training data, it contains 5153, 4739 and 4738 images for cat, dog and wild respectively. Each domain has 500 testing images. We performed Cat→Dog, Wild→Dog and multi-domain translation on this dataset.

During training, all images are resized 256×256, randomHorizontalFliped with p = 0.5, and scaled to $[-1, 1]$. During sampling, all images are resized 256×256 and scaled to $[-1, 1]$.

### B.2  Code Used and License

All used codes in this paper and its license are listed in Table 1.

### B.3  Details of the Score-based Diffusion Model

On Cat→Dog and Wild→Dog, we use the public pre-trained score-based diffusion model (SBDM) provided in the official code https://github.com/jychoi118/ilvr_adm of ILVR [2]. The pretrained model includes the variance and mean networks and we only use the mean network.

On Male → Female, we trained a SBDM for $1M$ iterations on the training set of female category using the recommended training code by SDEdit https://github.com/ermongroup/ddim. We use the same setting as SDEdit [9] and DDIM [11] for a fair comparison, where the models is trained with a

Table 2: Computation cost comparison.

| Methods | sec/iter↓ | Mem(GB)↓ |
|---------|-----------|----------|
| CUT     | 0.24      | 2.91     |
| ILVR    | 60        | 1.84     |
| SDEdit  | 33        | 2.20     |
| EGSDE   | 85        | 3.64     |

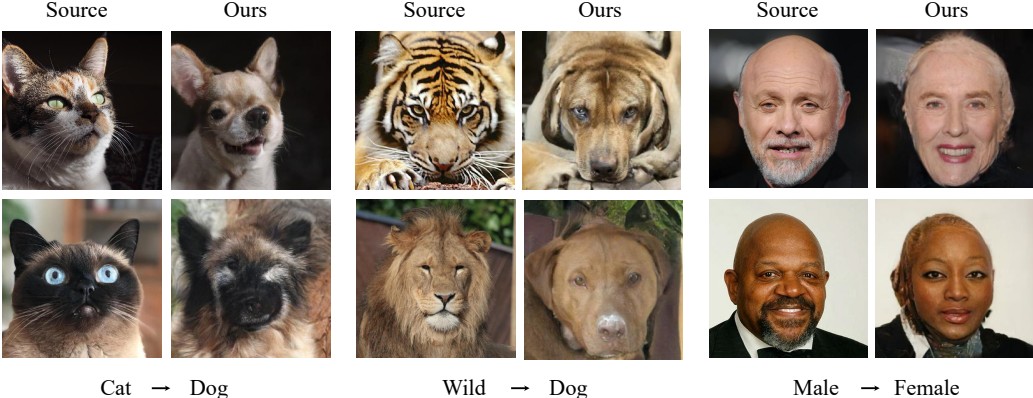

Figure 2: Selected failure cases. On Cat → Dog, the EGSDE sometimes fails to generate eyes and noses. On Wild → Dog, the EGSDE sometimes preserves some undesired features of the source image like tiger stripes. On Male → Female, the EGSDE fails to change the hairstyle.

batch size of $64$, a learning rate of $0.00002$, the Adam optimizer with $\beta_1 = 0.9$, $\beta_2 = 0.999$ and grad clip = 1.0, an exponential moving average (EMA) with a rate of $0.9999$. The U-Net architecture is the same as [7]. The timesteps $N$ is 1000 and the noise schedule is linear as described in A.3.

## B.4 Details of the Domain-specific Feature Extractor

The domain-specific feature extractor $E_s(\cdot, \cdot)$ is the all but the last layer of a classifier that is trained on both the source and target domains. The time-dependent classifier is trained using the official code https://github.com/openai/guided-diffusion of [4]. We use the ImageNet ($256\times256$) pretrained classifier provided in https://github.com/openai/guided-diffusion as the initial weight and train $5K$ iterations for two-domain I2I and $10K$ iterations for multi-domain I2I. We train the classifier with a batch size of 32, a learning rate of $3e - 4$ with the AdamW optimizer (weight decay = 0.05). For the architecture, the depth is set to 2, the channels is set to 128, the attention resolutions is set to 32,16,8 and the other hyperparameters are the default setting. The timesteps $N$ is 1000 and the noise schedule is linear.

## B.5 Training and Inference Time

On Cat → Dog, training the domain-specific feature extractor for $5K$ iterations takes 7 hours based on 5 2080Ti GPUs. The computation cost comparison for sampling is shown in Table 2, where the batch size is set 1. Compared with the ILVR, ours takes 1.42 times as long as ILVR. The speed of inference can be improved further by the latest progress on faster sampling [11, 1].

## B.6 Evaluation

**FID**. We evaluate the FID metric using the code https://github.com/mseitzer/pytorch-fid. On AFHQ dataset, following CUT [10], we use the test data as reference without any data preprocessing. On CelebA-HQ dataset, following StarGANv2 [10], we use the training data as reference and conduct the following data preprocessing: resize images to 256, 299 and then normalize data with

Table 3: The results of different similarity metrics. NS $L_2$ denote negative square $L_2$ distance. Cosine similarity for $\mathcal{S}_s$ and NS $L_2$ for $\mathcal{S}_i$ is the default setting.

| $\mathcal{S}_s$ | $\mathcal{S}_i$ | $\lambda_s$ | $\lambda_i$ | FID $\downarrow$ | L2 $\downarrow$ | PSNR $\uparrow$ | SSIM $\uparrow$ |
|---|---|---|---|---|---|---|---|
| Cosine | Cosine | 500 | 500 | 61.47 | 52.16 | 18.69 | 0.407 |
| Cosine | Cosine | 500 | 10000 | 64.23 | 50.97 | 18.89 | 0.408 |
| NS $L_2$ | NS $L_2$ | 5e-07 | 2 | 77.01 | 45.91 | 19.84 | 0.431 |
| NS $L_2$ | NS $L_2$ | 5e-05 | 2 | 65.90 | 51.05 | 18.89 | 0.403 |
| Cosine | NS $L_2$ | 500 | 2 | 65.23 | 47.15 | 19.32 | 0.415 |
| Cosine | NS $L_2$ | 500 | 1 | 63.78 | 47.88 | 19.19 | 0.413 |

$mean = 0.485, 0.456, 0.406, std = 0.229, 0.224, 0.225$. Note that the FID evaluation in StarGANv2 is still different with ours because it generates 10 images for each source image.

**Human evaluation.** We evaluate the human preference from both faithfulness and realism aspects via the Amazon Mechanical Turk (AMT). Given a source image, the AMT workers are instructed to select which translated image is more satisfactory in the pairwise comparisons between the baselines and EGSDE. The reward for each pair of picture comparison is kept as 0.02$. Since each task takes around 4 s, the wage is around 18$ per hour.

### B.7 Reproductions

All baselines are reproduced based on the public code. Specifically, CUT [10] is reproduced based on the official code https://github.com/taesungp/contrastive-unpaired-translation. On Cat→Dog, we use the public pretrained model directly without training. Following the setting on Cat→Dog, we train the CUT $2M$ iterations for other tasks. ILVR [2] is reproduced using the official code https://github.com/jychoi118/ilvr_adm. The diffusion steps is set to 1000. The $down\_N$ of low-pass filter is set to 32. The $range\_t$ is set to 20. SDEdit [9] is reproduced using the official code https://github.com/ermongroup/SDEdit, where we use the default setting. For StarGANv2, we use the public checkpoint in https://github.com/clovaai/stargan-v2 for evaluation.

## C Ablation Studies

### C.1 Choice of the Similarity Metrics

In this section, we perform two popular similarity metrics: cosine similarity and negative squared $L_2$ distance, for the similarity function $\mathcal{S}_s(\cdot, \cdot, \cdot)$ and $\mathcal{S}_i(\cdot, \cdot, \cdot)$. As shown in Table 3, the cosine similarity for $\mathcal{S}_s(\cdot, \cdot, \cdot)$ helps to improve the FID score notably and the negative squared $L_2$ distance helps to preserve more domain-independent features of the source image empirically, which are used finally in our experiments.

### C.2 An Alternative of Energy Function

In this section, we consider a simpler energy function that only involves the original source image $\boldsymbol{x}$ as follows:

$$\mathcal{E}(\boldsymbol{y}, \boldsymbol{x}, t) = \lambda_s \mathcal{S}_s(\boldsymbol{y}, \boldsymbol{x}, t) - \lambda_i \mathcal{S}_i(\boldsymbol{y}, \boldsymbol{x}, t), \tag{17}$$

which does not require to take the expectation w.r.t. $\boldsymbol{x}_t$. As shown in Table 4, it did not perform well because it is not reasonable to measure the similarity between the noise-free source image and the transferred sample in a gradual denoising process.

### C.3 Choice of Initial Time $M$

In this section, we explore the effect of the initial time $M$. The quantitative results are shown in Table 5. We found that the larger $M$ results in more realistic and less faithful images, because it preserve less information of the source image at start time with the increase of $M$.

Table 4: The results of different energy function. Variant denotes the choice of simpler energy function. The experiments are repeated 5 times to eliminate randomness.

| Model | FID ↓ | L2 ↓ | PSNR ↑ | SSIM ↑ |
|---|---|---|---|---|
| | | Cat → Dog | | |
| Variant | 79.01 ± 0.92 | 55.95 ± 0.06 | 17.86 ± 0.01 | 0.369 ± 0.000 |
| EGSDE | **65.82 ± 0.77** | **47.22 ± 0.08** | **19.31 ± 0.02** | 0.415 ± 0.001 |
| | | Wild → Dog | | |
| Variant | 67.87 ± 0.99 | 60.32 ± 0.05 | 17.23 ± 0.01 | 0.325 ± 0.001 |
| EGSDE | **59.75 ± 0.62** | **54.34 ± 0.08** | **18.14 ± 0.01** | **0.343 ± 0.001** |
| | | Male → Female | | |
| Variant | **41.86 ± 0.36** | 56.18 ± 0.03 | 17.89 ± 0.01 | 0.494 ± 0.000 |
| EGSDE | 41.93 ± 0.11 | **42.04 ± 0.03** | **20.35 ± 0.01** | **0.574 ± 0.000** |

Table 5: The results of different initial time $M$. The larger $M$ results in more realistic and less faithful images.

| Initial Time $M$ | FID ↓ | L2 ↓ | PSNR ↑ | SSIM ↑ |
|---|---|---|---|---|
| | | Cat → Dog | | |
| 0.3T | 97.02 | 33.39 | 22.17 | 0.516 |
| 0.4T | 78.64 | 39.95 | 20.70 | 0.461 |
| 0.5T | 65.23 | 47.15 | 19.32 | 0.415 |
| 0.6T | 57.31 | 55.98 | 17.88 | 0.374 |
| 0.7T | 53.01 | 65.61 | 16.55 | 0.333 |
| | | Wild → Dog | | |
| 0.3T | 96.80 | 38.76 | 20.93 | 0.472 |
| 0.4T | 73.86 | 46.50 | 19.43 | 0.395 |
| 0.5T | 58.82 | 54.34 | 18.14 | 0.344 |
| 0.6T | 55.53 | 62.52 | 16.94 | 0.307 |
| 0.7T | 54.56 | 72.02 | 15.72 | 0.274 |
| | | Male → Female | | |
| 0.3T | 51.66 | 31.66 | 22.71 | 0.639 |
| 0.4T | 47.13 | 36.74 | 21.48 | 0.605 |
| 0.5T | 42.09 | 42.03 | 20.35 | 0.574 |
| 0.6T | 36.07 | 48.94 | 19.09 | 0.534 |
| 0.7T | 30.59 | 59.18 | 17.48 | 0.472 |

Table 6: Comparison with SDEdit [9] under different $K$ times.

| Methods | K | Wild $\to$ Dog | | | | Cat $\to$ Dog | | | |
| --- | --- | --- | --- | --- | --- | --- | --- | --- | --- |
| | | FID↓ | L2↓ | PSNR↑ | SSIM↑ | FID↓ | L2↓ | PSNR↑ | SSIM↑ |
| SDEdit [9] | 1 | 68.22 | 55.38 | 17.97 | **0.342** | 73.70 | 47.74 | 19.22 | **0.424** |
| EGSDE | | **58.85** | **54.38** | **18.13** | 0.342 | **66.34** | **47.20** | **19.30** | 0.415 |
| SDEdit [9] | 2 | 60.91 | 62.32 | 16.97 | 0.312 | 65.59 | 55.10 | 18.01 | **0.395** |
| EGSDE | | **55.47** | **60.25** | **17.28** | **0.314** | **62.23** | **53.45** | **18.26** | 0.385 |
| SDEdit [9] | 3 | 60.52 | 66.16 | 16.46 | 0.303 | 61.10 | 59.69 | 17.33 | **0.382** |
| EGSDE | | **55.07** | **63.15** | **16.86** | **0.304** | **59.78** | **56.41** | **17.81** | 0.376 |

Table 7: The results of different $\lambda_s$ and $\lambda_i$. $\lambda_s = \lambda_i = 0$ corresponds to SDEdit [9].

| $\lambda_s, \lambda_i$ | Cat $\to$ Dog | | | | Male $\to$ Female | | | |
| --- | --- | --- | --- | --- | --- | --- | --- | --- |
| | FID↓ | L2↓ | PSNR↑ | SSIM ↑ | FID↓ | L2↓ | PSNR ↑ | SSIM↑ |
| $\lambda_s = 0, \lambda_i = 0$ | 73.85 | 47.87 | 19.19 | 0.423 | 49.68 | 43.68 | 20.03 | 0.572 |
| $\lambda_s = 100, \lambda_i = 0$ | 66.17 | 48.56 | 19.07 | 0.419 | 44.97 | 44.26 | 19.92 | 0.569 |
| $\lambda_s = 500, \lambda_i = 0$ | 62.44 | 51.02 | 18.64 | 0.405 | 38.44 | 45.92 | 19.6 | 0.559 |
| $\lambda_s = 800, \lambda_i = 0$ | 60.14 | 52.92 | 18.33 | 0.397 | 36.14 | 47.05 | 19.39 | 0.551 |
| $\lambda_s = 0, \lambda_i = 0.5$ | 74.09 | 45.58 | 19.61 | 0.428 | 50.77 | 41.67 | 20.43 | 0.58 |
| $\lambda_s = 0, \lambda_i = 2$ | 77.05 | 44.23 | 19.86 | 0.431 | 51.42 | 40.29 | 20.71 | 0.585 |
| $\lambda_s = 0, \lambda_i = 5$ | 79.12 | 43.63 | 19.98 | 0.433 | 52.13 | 39.57 | 20.87 | 0.588 |

## C.4 Repeating $K$ Times

In this section, we provide the comparison with SDEdit [9] under different $K$ times on Cat $\to$ Dog and Wild $\to$ Dog. The experimental results are reported in Table 6 and it is consistent with the results in the main text on Male $\to$ Female. With the increase of $K$, the faithful metrics of SDEdit decrease sharply, because it only utilizes the source image at the initial time $M$.

## C.5 Choice of $\lambda_s$ and $\lambda_i$

In this section, we provide the effect of weighting hyper-parameter $\lambda_s$ and $\lambda_i$ on Cat $\to$ Dog and Male $\to$ Female. The results are shown in Table 7 and Figure 3. It is consistent with the results in the full text on Wild $\to$ Dog. Larger $\lambda_s$ results in more realistic images and larger $\lambda_i$ results in more faithful images.

## C.6 More Qualitative Results

In this section, we show more qualitative results on three unpaired I2I tasks using the default hyper-parameters in Figure 1. We also select some failure cases in Figure 2 and randomly selected qualitative results in Figure 4.

## C.7 Comparison with StarGAN v2

In this section, we compare the EGSDE with StarGAN v2[3] on the most popular benchmark $Cat \to Dog$. Since the FID measurement in CUT we mainly follow and StarGAN v2 is different, for fairness, we perform experiments under both FID metrics. The results are shown in Table 1 and Table 8. The qualitative comparisons are shown in Figure 5.

As shown in Table 1 and Table 8, the EGSDE outperforms StarGAN v2 in all metrics under the two different measurements. It also should be noted that the three faithful metrics for StarGAN v2 is

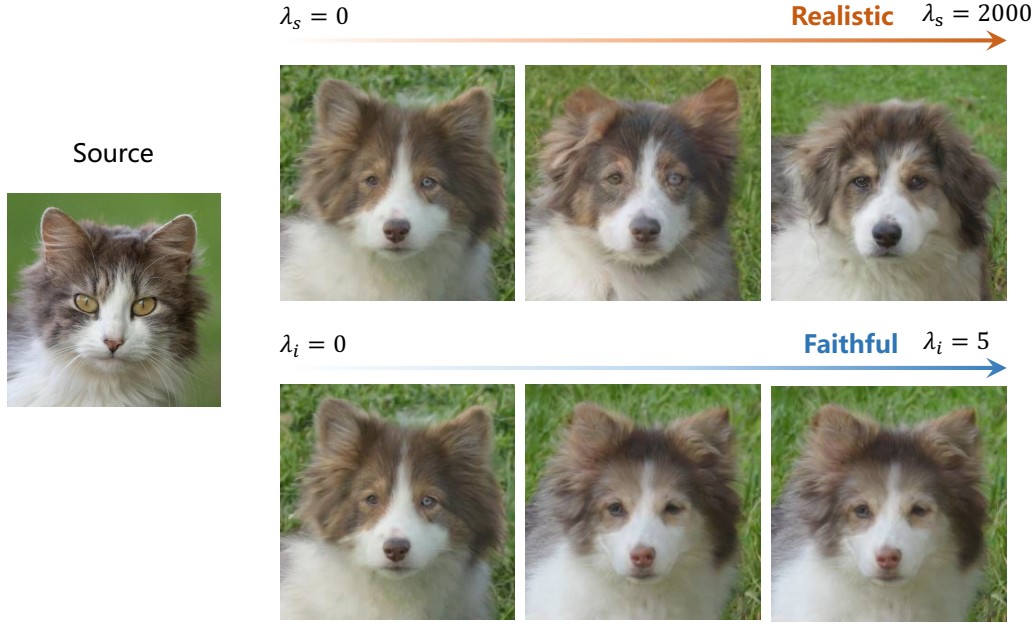

Figure 3: The qualitative results about the ablation of experts.

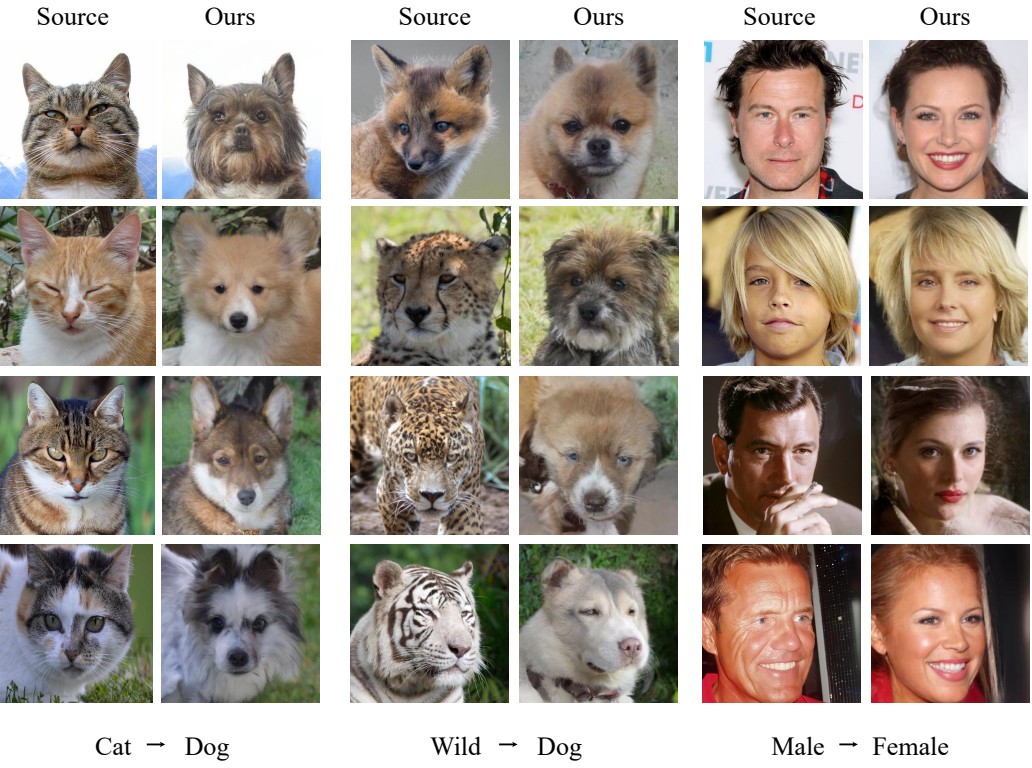

Cat → Dog          Wild → Dog          Male → Female

Figure 4: The randomly selected qualitative results with EGSDE.

Table 8: The comparison with StarGAN v2[3] on Cat $\rightarrow$ Dog following the FID measurement of StarGAN v2. The EGSDE use the default-parameters ($\lambda_s = 500, \lambda_i = 2, M = 0.5T$) and EGSDE$^{\dagger}$ use the parameters ($\lambda_s = 700, \lambda_i = 10, M = 0.6T$).

| Methods | FID$\downarrow$ |
|---|---|
| StarGAN v2 [3] | 36.37 |
| EGSDE | 48.20 |
| EGSDE$^{\dagger}$ | **31.14** |

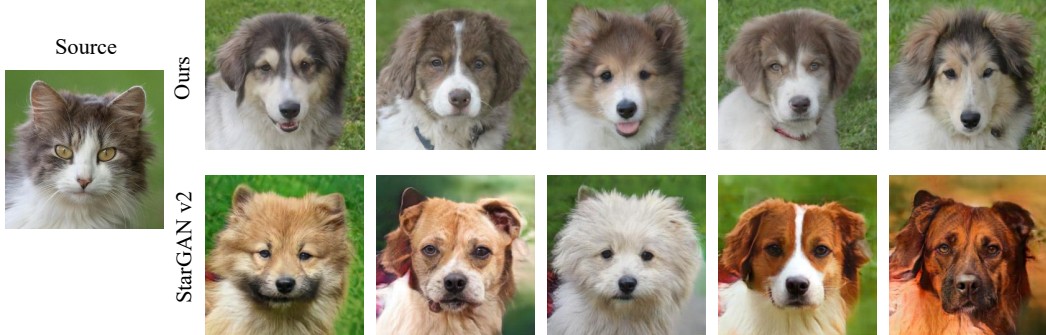

Figure 5: The qualitative comparisons with StarGAN v2. Each method generates five images by random seed for each source image. StarGAN v2 loses much domain-independent features (e.g. background and color) while EGSDE can still retain them.

much worse than ours. This is because the goal of StarGAN v2 is to generate diverse images over multi-domains, which pays little attention into the faithfulness. The qualitative comparisons in Figure 5 shows that StarGAN v2 loses much domain-independent features (e.g. background and color) while EGSDE can still retain them.

## D  Multi-Domain Image Translation

Following [2], we extend our method into multi-domain translation on AFHQ dataset, where the source domain includes *Cat* and *Wild* and the target domain is *Dog*. In this setting, similar to two-domain unpaired I2I, the EGSDE also employs an energy function pretrained on both the source and target domains to guide the inference process of a pretrained SDE. The only difference is the domain-specific feature extractor $E_s(,)$ involved in the energy function is the all but the last layer of a three-class classifier rather than two-class. All experiments are repeated 5 times to eliminate randomness. The quantitative results are reported in Table 11. We can observe that the EGSDE outperforms the baselines in almost all realism and faithfulness metrics, showing the great generalization of our method.

Table 9: The results of replacing $E_s$ with classifier guidance on Cat $\rightarrow$ Dog.

| Methods | FID$\downarrow$ | L2$\downarrow$ | PSNR$\uparrow$ | SSIM $\uparrow$ |
|---|---|---|---|---|
| EGSDE ($\lambda_s = 500, \lambda_i = 2$) | 65.82 | 47.22 | 19.31 | 0.415 |
| EGSDE-Classifier($\lambda_s = 5, \lambda_i = 2$ ) | 73.36 | 46.4 | 19.75 | 0.430 |
| EGSDE-Classifier ($\lambda_s = 50, \lambda_i = 2$) | 71.90 | 46.8 | 19.67 | 0.428 |
| EGSDE-Classifier ($\lambda_s = 500, \lambda_i = 2$) | 68.80 | 47.89 | 19.46 | 0.423 |

Table 10: The comparison with EGSDE-DDIM.

| Methods | FID$\downarrow$ | L2$\downarrow$ | PSNR$\uparrow$ | SSIM $\uparrow$ |
|---|---|---|---|---|
| EGSDE-DDPM($\lambda_s = 0, \lambda_i = 0$) | 74.17 | 47.88 | 19.19 | 0.423 |
| EGSDE-DDPM($\lambda_s = 500, \lambda_i = 2$) | 65.82 | 47.22 | 19.31 | 0.415 |
| EGSDE-DDPM($\lambda_s = 500, \lambda_i = 0$) | 62.44 | 51.02 | 18.64 | 0.405 |
| EGSDE-DDPM($\lambda_s = 0, \lambda_i = 2$) | 77.05 | 44.23 | 19.86 | 0.431 |
| EGSDE-DDIM($\lambda_s = 0, \lambda_i = 0$) | 88.29 | 41.93 | 20.62 | 0.472 |
| EGSDE-DDIM($\lambda_s = 500, \lambda_i = 2$) | 78.11 | 41.95 | 20.61 | 0.468 |
| EGSDE-DDIM($\lambda_s = 500, \lambda_i = 0$) | 74.32 | 44.32 | 20.12 | 0.460 |
| EGSDE-DDIM($\lambda_s = 0, \lambda_i = 2$) | 91.81 | 39.80 | 21.10 | 0.481 |

Table 11: Quantitative results in multi-domain translation, where the source domain includes *Cat* and *Wild* and the target domain is *Dog*. All experiments are repeated 5 times to eliminate randomness.

| Methods | FID$\downarrow$ | L2$\downarrow$ | PSNR$\uparrow$ | SSIM $\uparrow$ |
|---|---|---|---|---|
| ILVR [2] | 74.85 ± 1.24 | 60.16 ± 0.14 | 17.31 ± 0.02 | 0.325 ± 0.001 |
| SDEdit [9] | 71.34 ± 0.64 | 51.62 ± 0.05 | 18.58 ± 0.01 | **0.383 ± 0.001** |
| EGSDE | **64.02 ± 0.43** | **50.74 ± 0.04** | **18.73 ± 0.01** | 0.373 ± 0.000 |