# OpenReview forum: "EGSDE: Unpaired Image-to-Image Translation via Energy-Guided Stochastic Differential Equations"
_NeurIPS.cc/2022/Conference — NeurIPS 2022 Accept_

### Official Review · Reviewer_3nHW · 2022-07-10

**Rating:** 6
**Confidence:** 3
**Soundness:** 3 good
**Presentation:** 3 good
**Contribution:** 3 good

**Summary:**

This paper studies SDGMs for image2image translation. The authors propose energy functions pretrained from both source and target dataset to guide the SDE inference process, to generate realistic and faithful images. Experiments show that the proposed method outperforms SDGM-based methods quantitatively.

**Questions:**

1.  What is the drift coefficient f(y,t) in practice?
2. Is sampling a start point $q_{M|0}(y_M|x_0))$ from $x_0$ still necessary, since a faithful expert is introduced? Start point and the faithful expert, who is the main contributor to faithfulness?
3. How is the computation cost of EGSDE, compared with SDGM-based baselines and GAN-based methods? Are the qualitative results elaborately picked or randomly selected?
4. I suggest the authors to present some qualitative results about the ablation of experts and start point.

**Limitations:**

A limitation is that, unlike the realistic expert,  the faithful expert simply uses a low-pass filter without knowledge of source and target datasets. As said in the paper, this could be improved in future work.

**Strengths And Weaknesses:**

Strengths:
1. Introducing the source domain knowledge to improve SDGM-based methods is promising and a good contribution.
2. Quantitative experimental results are rich. The selection of baselines and evaluation metrics is reasonable.
3. The paper is overall well written.

Weaknesses:
1. The background of SDGMs is a bit hard to follow. I suggest the authors give an intuitive understanding of what each equation does for image2image.  Also, I can't find what is f(y,t) in practice.
2. The paper lacks some qualitative analysis about the ablation of each expert.

---

> ### Author Response · Authors · 2022-08-02
> **Author Response to Reviewer 3nHW**
>
> We thank reviewer 3nHW for the valuable and constructive comments.
> ## Q1: Give an intuitive understanding of each equation  in the background of SDGMs and the meaning of f(y,t) in practice.
> Thank you for the suggestion. We add some intuitive understanding of each equation in the background of SDGMs. The Eq. (1) describes how to add noise in the forward diffusion process. The $f(y,t),g(t)$ is related into the noise size and determine the perturbation kernel $q_{t|0}(y_t|y_0)$ (see line 62 in the revision). In practice, the $f(y,t)$ is usually affine ($f(y,t)=y t$) to make sure the the perturbation kernel is a linear Gaussian distribution and can be sampled in one step (see line 63 in the updated version). For example, in the VP-SDE (DDPM [4] is the discretization of it), $f(y,t) = -\frac{1}{2}\beta(t) y, g(t) = \sqrt{\beta(t)}$. The Eq. (2) describe the reverse denoising process we wish to solve, where $f(y,t),g(t)$ has been given by the forward process and we only need to learn the true score function $\nabla_y \log q_t(y)$ (see line 68 in the updated version). The Eq. (3) means if we have a score model $s(y,t)$ learned by score-matching methods, then we can replace the true score function in the Eq. (2) with it to generate images. The solution of Eq. (3) is corresponding to the generative model, where the $f(y,t),g(t)$ and $s(y,t)$ is known. We can use various SDE solver (e.g. Euler-Maruyama solver) to solve it to generate images, which is the meaning of Eq. (4) (see line 72 in the revision).
>
> ## Q2: Is sampling a start point still necessary, since a faithful expert is introduced? Start point and the faithful expert, who is the main contributor to faithfulness?
> Thank you for the suggestion. We add the experiment which drop the start point and faithful expert respectively in Rebuttal Table 4. We can observe dropping the start point leads to more decrease in the faithful metrics. Therefore, the start point is the main contributor to faithfulness and it's necessary.
>
> **Rebuttal Table 4.** The faithful comparison of results with dropping the start point and faithful expert.
> |Method | L2$\downarrow$      | PSNR$\uparrow$    | SSIM $\uparrow$   |
> | :----: | :----: | :----: | :----: |
> |EGSDE  | 47.22 | 19.31  | 0.415  |
> |dropping start point  | 82.51  |  14.72 | 0.270  |
> |dropping faithful expert | 51.02  |  18.64 | 18.64  |
> | | | | |
>
> ## Q3: The computation cost of EGSDE and other baselines.
> Thank you for the suggestion. As suggested, we add the computation cost comparison in Rebuttal Table 5, where the batch size is set 1. Compared with the ILVR, ours takes 1.42 times as long as ILVR. We also add this results in Table 11 in Appendix in the revision.
>
> **Rebuttal Table 5.** Computation cost comparison.
> |Method | sec/batch $\downarrow$      | Mem (GB) $\downarrow$     |
> | :----: | :----: | :----: |
> | CUT |  0.24 |  2.91  |
> | ILVR  |  60 |  1.84  |
> | SDEdit |  33 |  2.20  |
> | EGSDE |  85 |  3.64  |
> | | | |
>
> ## Q4: Are the qualitative results elaborately picked or randomly selected?
> We show the representative translation results in Figure 1, Figure 3 and Figure 1 in Appendix.  We show more randomly selected qualitative results in Figure 5 in the Appendix and we also select some failure cases in Figure 2 the Appendix.
>
> ## Q5: Add qualitative results about the ablation of experts and start point.
> (1) Thank you for the suggestion. As for the ablation of experts, we showed one example in Figure 1 in the submission version and we add more qualitative results in Figure 6 in Appendix. The qualitative results are consistent with the quantitative results, where larger $\lambda_s$ results in more realistic images and larger $\lambda_i$ results in more faithful images, which support our motivation.
>
> (2) We did the ablation studies of start point in the submission version on three tasks. The quantitative and qualitative results are shown in Table 4 and Figure 3 in Appendix C.3.  We can observe the larger M results in more realistic and less faithful images, because it preserves less information of the source image at start time with the increase of M.
>
> [4] Jonathan Ho, Ajay Jain, and Pieter Abbeel. Denoising diffusion probabilistic models. Advances in Neural Information Processing Systems, 33:6840–6851, 2020.

---

> > ### Comment · Reviewer_3nHW · 2022-08-09
> > **Response**
> >
> > Thank you for your response. The rebuttal addresses my concerns. I lean towards weak acceptance.

---

### Official Review · Reviewer_gtyS · 2022-07-10

**Rating:** 6
**Confidence:** 3
**Soundness:** 3 good
**Presentation:** 3 good
**Contribution:** 2 fair

**Summary:**

The paper proposes Score-based diffusion generative models (SDGMs) that does not ignore the training data in the source domain, leading to improved solutions for unpaired I2I. The proposed method is  energy-guided stochastic differential equations (EGSDE) that employs an energy function pretrained on both the source and target domains to guide the inference process of a pretrained SDE for realistic and faithful unpaired I2I.

**Questions:**

None

**Limitations:**

No immediate negative impact

**Strengths And Weaknesses:**

**Weakness**
- Evaluation on more standard datasets is missing.

**Strengths**
- Intuitive idea, well-written paper.

---

> ### Author Response · Authors · 2022-08-02
> **Author Response to Reviewer gtyS**
>
> We thank the reviewer gtyS for the acknowledgement of our contributions.

---

### Official Review · Reviewer_WJZs · 2022-07-11

**Rating:** 6
**Confidence:** 4
**Soundness:** 3 good
**Presentation:** 3 good
**Contribution:** 3 good

**Summary:**

The author proposes an unpaired image-to-image translation method using score-based diffusion generative models. Compared with existing arts, the method also considers the training data in the source domain. To be specific, the method deploys an energy function pre-trained across two image domains. Experiments show the superiority of the proposed method.

**Questions:**

Questions:
1. Please give the detailed implementation of E_i.
2. How about denoising from a perturbed image using DDIM reverse sampling instead of denoising from random noise?


**Limitations:**

The limitations and potential negative societal impact have been adequately addressed.

**Strengths And Weaknesses:**

Strength:
1. This paper considers the domain-independent and domain-dependent features in diffusion models, which is not explored until recently.
2. Measure content similarity with energy function in the diffusion model is a good idea. The authors also provide a theoretical insight into EGSDE.
3. The comparisons are sound and comprehensive.

Weaknesses:
1. One state-of-the-art image translation is missing. Please compare EGSDE with DiffusionCLIP [1].
2. There are many other ways to address image translation and content preserving. One can use classifier guidance to generate images in the target domain [2]. To preserve identity information, one can use identity features. The author should highlight the novelty or the advancement compared with these solutions.
3. The paper will benefit from a more exhaustive ablation study. For example 1) the effectiveness of E_s and E_i. 2) replacing E_s with classifier guidance [2] or classifier-free guidance [3].
4. The paper will benefit from a human evaluation. i.e., comparing EGSDE with SDEdit and ILVR on Male -> Female.

[1] DiffusionCLIP: Text-Guided Diffusion Models for Robust Image Manipulation
[2] Diffusion Models Beat GANS on Image Synthesis
[3] Classifier-Free Diffusion Guidance

---

> ### Author Response · Authors · 2022-08-02
> **Author Response to Reviewer WJZs**
>
> We thank the reviewer WJZs for the valuable  comments.
> ## Q1: Comparison with DiffusionCLIP
> Thank you for the suggestion. DiffusionCLIP is a novel SDGMs-based method for image manipulation and has achieved great performance in a wide range of applications including text-driven image manipulation, zero-shot image manipulation between unseen domains and multi-attribute transfer. We also notice that DiffusionCLIP fine-tunes the noise prediction network $\epsilon_\theta$ with CLIP loss, which uses the additional pretrained CLIP model and text data. This is incomparable with unparied image-to-timage translation tasks in this paper. We really thank the reviewer for pointing out the missing of this important related work. We add the discussion about DiffusionCLIP in Section 4 in the revision (see line 204-207).
>
> ## Q2: Highlight the novelty compared with the existing classifier guidance.
> Our main contribution is not designing an energy function form but proposing a general framework EGSDE that employs an energy function with domain knowledge to guide the inference process and providing an explanation of EGSDE as a product of experts. The design of energy function is various. In this work, we provide a simple and intuitive design of energy function for unpaired I2I task and achieve good results. The classifier guidance can also be regarded as a special design of energy function. If more sophisticated and better energy function could be designed, the performance could be improved further.
>
> ## Q3: Add the ablation studies for $E_s$ and $E_i$ and replace $E_s$ with classifier guidance.
> (1) Thank you for the suggestion. We did the the ablation studies for $ E_s$ and $E_i$ in the original submission version, where the results on Wild $\to$ Dog are reported in Table 3 and the results on other tasks are reported in Table 6 in Appendix. The experiments show larger $\lambda_s$ results in more realistic images and larger $\lambda_i$ results in more faithful images, which support our motivation.
>
> (2) As suggested, we add the experiments for EGSDE-Classifier, where the $E_s$ is replaced with classifier guidance on Cat $\rightarrow$ Dog in Rebuttal Table 3. We also add this experiments in Table 10 in Appendix.
>
> **Rebuttal Table 3.** The results of replacing $E_s$ with classifier guidance on Cat $\rightarrow$  Dog.
> |Method | FID$\downarrow$     | L2$\downarrow$      | PSNR$\uparrow$    | SSIM $\uparrow$   |
> | :----: | :----: | :----: | :----: | :----: |
> | EGSDE ($\lambda_s = 500, \lambda_i = 2$)  |  65.82  |  47.22  |  19.31  |  0.415 |
> | EGSDE-Classifier($\lambda_s = 5, \lambda_i = 2$ )  | 73.36 |  46.4 |  19.75 | 0.430  |
> | EGSDE-Classifier ($\lambda_s = 50, \lambda_i = 2$)  |  71.90 |  46.8 | 19.67 |  0.428  |
> | EGSDE-Classifier ($\lambda_s = 500, \lambda_i = 2$)  |  68.80 |  47.89 | 19.46 |  0.423  |
> | | | | | |
>
> ## Q4: Add human evaluation. i.e., comparing EGSDE with SDEdit and ILVR on Male $\rightarrow$ Female.
> Thank you for the suggestion. We add human evaluation as suggested. More details are available in our response to the Common Concern, where EGSDE is preferred with 74.4% and 88.2% compared with SDEdit and ILVR respectively on Male $\rightarrow$ Female.
>
> ## Q5: The detailed implementation of $E_i$
> Thank you for the suggestion. We use the resize function including downsampling and upsampling operation as low-pass filter. We add this details in Section 5 Experiments (see line 219).
>
> ## Q6: How about denoising from a perturbed image using DDIM reverse sampling?
> We add the experiments based on DDIM in Rebuttal Table 6. The results show the EGSDE can also be validated using DDIM , where ours improves realism results without harming the faithful performance. The ablation of each expert implys the realistic expert helps to discard domain-specific features and the faithful expert helps to preserve the domain-independent ones. We also note that compare with EGSDE-DDPM under same hyper-parameters, the EGSDE-DDIM tend to have higher FID and lower $L_2$ distance. We add the above results in Table 12 in Appendix.
>
> **Rebuttal Table 6.** The comparison with EGSDE-DDIM.
> |Method | FID$\downarrow$    | L2$\downarrow$     | PSNR$\uparrow$   | SSIM $\uparrow$ |
> | :----: | :----: | :----: | :----: | :----: |
> | EGSDE-DDPM($\lambda_s = 0, \lambda_i=0$)            |  74.17          |  47.88         |  19.19          |  0.423 |
> | EGSDE-DDPM($\lambda_s = 500, \lambda_i=2$) |  65.82 |  47.22 | 19.31  |  0.415|
> | EGSDE-DDPM($\lambda_s = 500, \lambda_i=0$) | 62.44 |  51.02  | 18.64 |  0.405|
> | EGSDE-DDPM($\lambda_s = 0, \lambda_i=2$) | 77.05 |  44.23   |  19.86 | 0.431 |
> | EGSDE-DDIM($\lambda_s = 0, \lambda_i=0$)    |  88.29 |  41.93 |  20.62  | 0.472|
> | EGSDE-DDIM($\lambda_s = 500, \lambda_i=2$)    | 78.11 |  41.95 |  20.61  |  0.468 |
> | EGSDE-DDIM($\lambda_s = 500, \lambda_i=0$)    |  74.32 |  44.32 |  20.12  |  0.460 |
> | EGSDE-DDIM($\lambda_s = 0, \lambda_i=2$)    |  91.81 &| 39.80 |  21.10  |  0.481 |
> | | | | | |

---

### Official Review · Reviewer_u4yM · 2022-07-11

**Rating:** 5
**Confidence:** 4
**Soundness:** 3 good
**Presentation:** 2 fair
**Contribution:** 2 fair

**Summary:**

This work introduced an energy-guided stochastic differential equation based method for Image2Image translation task. Unlike previous methods, the proposed one takes the source training data into account and designs the energy function to preserve the domain-independent features and discard domain-specific features. Experiments on faces and animal faces show the effectiveness of the proposed method over existing GAN based and diffusion based I2I works.

**Questions:**

Check my concerns above.

**Limitations:**

Yes.

**Strengths And Weaknesses:**

+ A nice interpretation of the discretization of EGSDE in the formulation of product of experts

Please find my concerns below:

(i) My biggest concern is that I’m not sure if energy or diffusion based models have beaten GAN based approaches, as claimed by authors. The most typical GAN based approach is StarGAN [8], which is cited by authors and the dataset source used in this work. However, I do not find any comparison with [8] in the experiment section? Any special reasons? Though [8] handles multi-domain translation, it is still okay to train the framework of [8] with just two domains. Additionally, the FID numbers shown in [8] are much lower than what is shown in this paper. Overall, I’m not quite convinced if GAN based approaches have been outperformed as they also take both source and target training data into consideration.

(ii) Is the proposed method based on energy function much slower than a feedforward step in GAN approach? If so, I probably expect even higher performance in terms of quality. User study is another good way to evaluate perceptual preference if there is no GT to evaluate.

(iii) It looks the EGSDE is not limited to I2I task. Would it be also applicable to other tasks like colorization or inpainting?


--------------
Thanks for the rebuttal from authors. After checking comments from other two reviewers, I increased my score to 5. The reason why I asked authors to compare with the GAN-based approach StarGAN v2 is that authors need to support the claim with solid experiments by saying the diffusion model has unique advantages on this task, not just because diffusion model is popular nowadays and has not much explored for I2I yet. Especially considering that authors have used the dataset from StarGAN work but eventually ignore the comparison in the draft. Be more rigorous and hopefully do not make this type of mistake again in future research.

---

> ### Author Response · Authors · 2022-08-02
> **Author Response to Reviewer u4yM**
>
> We thank the reviewer u4yM for the valuable and constructive comments.
> ## Q1: Comparison with StarGAN v2, where the FID is much lower than ours.
> (1) Following your suggestion, we add comparisons with StarGAN v2 [1] on the most popular benchmark Cat $\rightarrow$ Dog. The EGSDE outperforms StarGAN v2 in all metrics used in our original submission, as shown in Rebuttal Table 2. See experimental details in Appendix C.9 . It also should be noted that the three faithful metrics for StarGAN v2 is much worse than ours. This is because StarGAN v2 aims to generate diverse images and pays little attention into the faithfulness, which is different from our goal. The qualitative comparisons in Figure 4 of Appendix shows that StarGAN v2 loses much domain-independent features (e.g. background and color) while EGSDE can still retain them.
>
> (2) We note that the FID metric in StarGAN V2 is different from the one used in our submission and CUT. For fairness, we compare StarGAN V2 and ours using the StarGAN's protocol as well in Table 9 of Appendix. The conclusion remains the same. We use the public checkpoint of StarGAN v2, which can reproduce the results reported in StarGAN v2 [1].
>
> **Rebuttal Table 2.** The comparison with StarGAN v2 on Cat $\rightarrow$ Dog.
> |Method | FID$\downarrow$    | L2$\downarrow$     | PSNR$\uparrow$   | SSIM $\uparrow$  |
> | :----: | :----: | :----: | :----: | :----: |
> |StarGAN v2                   |54.88 ± 1.01               |133.65 ± 1.54                | 10.63 ± 0.10               | 0.27 ± 0.003  |
> |EGSDE   | 51.04 ± 0.37 | 62.06 ± 0.10 | 17.17 ± 0.02  | 0.36 ± 0.001|
> | | | | | |
>
> ## Q2: EGSDE is much slower than GAN-based method and the reviewer expect higher performance.
> (1) Our motivation is to improve the existing SDGMs-based methods. As for efficiency, our method is comparable to the most direct competitors, i.e., SDGM-based methods. For instance, ours takes 1.42 times as long as ILVR. SDGMs-based methods are slower than GAN-based methods currently. However, the speed of such methods can be improved further by the recent advances on faster sampling methods [2,3], which achieve comparable results with VP-SDE (1000 steps) using only around 10 Steps.
>
> (2) In addition, we can further improve the FID by tuning hyper-parameters. We add experiments with $\lambda_s = 700, \lambda_i = 0.5, M = 0.6T$ in three tasks in Table 1 of the revision. Under this setting, we improve the FID to 51.04 and 50.43 on the Cat $\rightarrow$ Dog and Wild \$\rightarrow$ Dog tasks respectively, which outperform the popular GAN-based method CUT by 25.17 and 42.51 .
>
>
> ## Q3: Add user study
> Thank you for the suggestion. We add human evaluation as suggested. More details are available in our response to the Common Concern, where EGSDE is preferred compared to baselines (> 50%).
>
>
> ## Q4: Would EGSDE be applicable to other tasks like colorization or inpainting?
> Yes. EGSDE can be applied in other image translation tasks and the user need to design the energy function carefully for the task. Take colorization as an example, the energy function should retain the content of the source image by learning a content extractor and change the color of the source image by learning a color extractor.
>
> [1] Yunjey Choi, Youngjung Uh, Jaejun Yoo, and Jung-Woo Ha. Stargan v2: Diverse image synthesis for multiple domains. In Proceedings of the IEEE/CVF conference on computer vision and pattern recognition, pages 8188–8197, 2020.
>
> [2] Fan Bao, Chongxuan Li, Jun Zhu, and Bo Zhang. Analytic-dpm: an analytic estimate of the optimal reverse variance in diffusion probabilistic models. arXiv preprint arXiv:2201.06503, 2022.
>
> [3] Cheng Lu, Yuhao Zhou, Fan Bao, Jianfei Chen, Chongxuan Li, and Jun Zhu. Dpm-solver: A fast ode solver for diffusion probabilistic model sampling in around 10 steps. arXiv preprint arXiv:2206.00927, 2022.

---

> > ### Author Response · Authors · 2022-08-05
> > **Looking forward to further feedback**
> >
> > Dear Reviewer u4yM,
> >
> > Thank you again for the great efforts and valuable comments. We have carefully addressed the main concerns in detail. We hope you might find the response satisfactory (similar as the other reviewers). As the discussion phase is about to close, we are very much looking forward to hearing from you about any further feedback. We will be very happy to clarify further concerns (if any).
> >
> > Best, Authors

---

> > > ### Author Response · Authors · 2022-08-09
> > > **Thanks for the update**
> > >
> > > Dear Reviewer u4yM,
> > >
> > > Thank you very much for the update of rating from 4 to 5. We believe that the revision with a direct comparison between EGSDE and StarGAN shows the promise of EGSDE. We are glad to clarify further concerns (if any).
> > >
> > > Best, Authors

---

### Author Response · Authors · 2022-08-02
**Common Concerns from reviewers**

We address the common concerns here and post a point-to-point response to each reviewer as well.
## Common concern 1 (from reviewer u4yM and WJZs ): Human Evaluation
As suggested, we add human evaluation in Rebuttal Table 1 . We evaluate the human preference from both faithfulness and realism aspects via the Amazon Mechanical Turk (AMT). Given a source image, the AMT workers are instructed to select which translated image is more satisfactory in the pairwise comparisons between the baselines and EGSDE. The results show that EGSDE is preferred compared to all baselines (> 50%). We add the above results in Table 1 and the experiments section (see line 225, line 241) in the revision. More details about human evaluation is available in C.10 in Appendix.

**Rebuttal Table 1.** Human evaluation via Amazon Mechanical Turk (AMT). The reported percentage (%) show the preference rate of EGSDE against baselines.

|Method | Cat $\rightarrow$ Dog | Wild $\rightarrow$ Dog | Male $\rightarrow$ Female |
| :----: | :----: | :----: | :----: |
|CUT                    | 79.6% |  82.4%  | 58.6%  |
|ILVR     |   75.4%  |  73.4%  |  88.2% |
|SDEdit                   | 65.2%   | 57.2% | 74.4% |
| | | | |

---

### Author Response · Authors · 2022-08-02
**Summary of the revision**

We sincerely thank all reviewers for their valuable comments, which help to further improve the quality of our work. We have thoroughly addressed the detailed comments, and summarize the revision in the updated version as follows:
* We add human evaluation using Amazon Mechanical Turk (see Table 1, line 225, line 241 and C.10 in Appendix)
* We add more experiments with other hyper-parameters (see Table 1)
* We add quantitative comparison with StarGAN v2 on the the most popular benchmark $Cat \to Dog$ under two different FID measurements (see Table 1 and Table 8, Table 9 in Appendix)
* We add quantitative comparison with StarGAN v2 (see Figure 4 in Appendix)
* We add more discussion about DiffusionCLIP in the Section 4 (see line 204-207)
* We add the experiments with classifier guidance (see Table 9 in Appendix)
* We add the detailed implementation of low-pass filter in Section 5 (see line 219)
* We add the experiments with DDIM sampling (see Table 12 in Appendix)
* We add some intuitive understanding of each equation in the background of SDGMs (see Section 2.1)
*  We add the computation cost of all the methods (see Table 11 in Appendix)
*  We add random selected qualitative results of EGSDE (see Figure 5 in Appendix)
* We add qualitative results about the ablation of each expert (see Figure 6 in Appendix)

We hope you may find the response satisfactory. Please let us know if you have any further feedback.

---

### Author Response · Authors · 2022-08-05
**Looking forward to further feedback**

Dear AC and Reviewers,

Thank you again for the great efforts and valuable comments. We have carefully addressed the main concerns in detail. We hope you might find the response satisfactory. As the discussion phase is about to close, we are very much looking forward to hearing from you about any further feedback. We will be very happy to clarify any further concerns (if any).

Best, Authors

---

### Comment · Area_Chair_otTh · 2022-08-09
**gentle reminder**

Dear reviewers,

Thank you all for providing valuable comments. The authors have provided detailed responses to your comments. Has the response addressed your major concerns?

I would appreciate it a lot if you could reply to the authors’ responses soon as the deadline is approaching (Tues, Aug 9).

Best,

ACs

---

### Meta-Review · Area_Chair_otTh · 2022-08-28

**Recommendation:** Accept
**Confidence:** Certain

**Metareview:**

The paper proposes an unpaired image-to-image translation method based on score-based diffusion models. Compared to prior works [7, 29], the paper adds two energy functions pretrained on both the source and target domains in an expert-of-product framework. The paper has received positive reviews. Reviewers found the paper well-written, the idea intuitive, and the experimental results comprehensive.  The rebuttal further addressed the concerns regarding the user study, running time, and missing comparisons. The AC agreed with the reviewers’ consensus and recommended accepting the paper.



**Award:**

No

---

### Decision · Program_Chairs · 2022-09-14

Accept